

# Capabilities of optical and radar Earth observation data for up-scaling methane emissions linked to subsidence and permafrost degradation in sub-Arctic peatlands

Sofie Sjögersten[1], Martha Ledger[1], Matthias Siewert[2], Betsabé de la Barreda-Bautista[1,3], Andrew Sowter[4], David Gee[4], Giles
5 Foody[3], Doreen S. Boyd[3]

[1]University of Nottingham, School of Biosciences, College Road, Sutton Bonington, Loughborough, LE12 5RD, UK

[2]University of Umeå, Department of Ecology and Environmental Sciences, KB H4, Linnaeus väg 6, Umeå universitet, 901
87 Umeå, Sweden

[3]School of Geography, University of Nottingham, University Park, Nottingham, NG7 2RD, UK

[4]Terra Motion Ltd, Ingenuity Centre, Triumph Rd, Nottingham NG7 2TU, UK

*Correspondence to*: Sofie.Sjogersten@nottingham.ac.uk

**Abstract.** Permafrost thaw in Arctic regions is increasing methane ($CH_4$) emissions to the atmosphere but quantification of
such emissions is difficult given the large and remote areas impacted. Hence, Earth Observation (EO) data are critical for
assessing both permafrost thaw, associated ecosystem change, and increased $CH_4$ emissions. Often extrapolation from field
measurements using EO is the approach employed. However, there are key challenges to consider - that landscape $CH_4$
emissions result from a complex local-scale mixture of micro-topographies and vegetation types that support widely
differing $CH_4$ emissions and the difficulty in detecting the initial stages of permafrost degradation before vegetation
transitions have occurred. This study considers the use of a combination of ultra-high resolution unoccupied aerial vehicle
(UAV) data, together with Sentinel-1 and -2 data to extrapolate field measurements of $CH_4$ emissions from a set of
vegetation types which capture the local variation in vegetation on degrading palsa wetlands. We show that the ultra-high
resolution UAV data can map spatial variation in vegetation relevant to variation in $CH_4$ emissions and extrapolate these
across the wider landscape. We further show how Sentinel-1 and Sentinel-2 can be used. By way of a soft classification, and
simple correction of misclassification bias of a hard classification, the output vegetation mapping and subsequent
extrapolation of $CH_4$ emissions matched closely that generated using the UAV data. InSAR assessment of subsidence
together with the vegetation classification suggested that high subsidence rates of palsa wetland can be used to quantify areas
at risk of increased $CH_4$ emissions. We estimate that a transition of an area currently experiencing subsidence to fen type
vegetation are estimated to increase emissions from 116 kg $CH_4$ season$^{-1}$ from the 50 ha study area, to emissions as high as
6500 to 13000 kg $CH_4$ season$^{-1}$. The key outcome from this study is that a fusion of EO data types provides the ability to
estimate $CH_4$ emissions from large geographies covered by a fine mixture of vegetation types and vulnerable to transitioning



to CH$_4$ emitters in the near future. This points to an opportunity to measure and monitor CH$_4$ from the Arctic over space and time with confidence.

## 1 Introduction


The severe impact of climate heating in the Arctic has raised grave concerns of positive feedbacks from ecosystems that store substantial amounts of soil organic carbon in soils and peat deposits (Hugelius et al., 2020; Mishra et al., 2021; Sjöberg et al., 2020; Schneider Von Deimling et al., 2012; Chadburn et al., 2017; Grosse et al., 2016; Biskaborn et al., 2019). Although the overall carbon balance in permafrost and non-permafrost peatlands is similar (Olefeldt et al., 2012), permafrost

thaw increases methane (CH$_4$) emissions across a range of Arctic landscapes (Olefeldt et al., 2013). Any increase in emissions of CH$_4$ is of particular concern, as it is a powerful greenhouse gas in a global context, with Arctic ecosystems acting as strong emitters of CH$_4$ (Mcguire et al., 2009; Turetsky et al., 2020; Euskirchen et al., 2014; Maksyutov et al., 2010; Glagolev et al., 2011).

Palsa are elevated peat mounds with a frozen core formed in peatlands and are a common landform in the sporadic

and discontinuous zone of the permafrost region where palsa peatlands cover ca. 20% of the total permafrost region (Ballantyne C. K., 2018; Tarnocai et al., 2009). From a climate feedback perspective, these are critical systems as they are carbon (C) rich, with an estimated storage of 100 Gt C (based on data in (Tarnocai et al., 2009; Schuur and Abbott, 2011). Palsa peatland are dynamic systems that comprise of raised peatland plateaux, which can vary in size from tens of meters to kilometers in size, these plateaux are often surrounded by lower lying waterlogged areas or have borders with thermokarst

lakes (Ballantyne C. K., 2018; Zuidhoff and Kolstrup, 2000). As a result of the raised nature of palsa their surface tend to be relatively dry. However, this changes as the permafrost in these peatlands degrade and the peat surface subsides which, over time, results in waterlogging (Sjöberg et al., 2015; Olvmo et al., 2020). Such degradation is already being observed in many areas of the Arctic in response to the greater than average warming occurring in this region (Åkerman and Johansson, 2008; Sannel et al., 2016; Olvmo et al., 2020; De La Barreda-Bautista et al., 2022; Borge et al., 2017; Luoto and Seppälä, 2003;

Sannel and Kuhry, 2011). The loss of permafrost and switch to waterlogged conditions represents a profound change in how this ecosystem functions, driving increased CH$_4$ emissions which has been observed across many Arctic landscapes (Miglovets et al., 2021; Varner et al., 2022; Glagolev et al., 2011; Walter Anthony et al., 2016; Walter et al., 2006). To understand fully the magnitude of these feedbacks and their impacts on the global climate, it is imperative to both quantify current emissions and predict new areas that may become CH$_4$ emitters across Arctic landscapes undergoing permafrost

degradation.

Given the size of the areas under threat from warming and their remoteness, remote sensing techniques are necessary to detect when and where permafrost degradation and subsidence is occurring. Coupled with measures of CH$_4$ flux, this may offer an approach for understanding, estimating, and predicting CH$_4$ feedbacks at landscape scales across permafrost peatlands of the circumarctic region. Detecting year-to-year degradation of permafrost manifested as subsidence



is possible using the Advanced Pixel System using Intermittent Baseline Subset (ASPIS-InSAR) and other InSAR techniques (Sowter et al., 2016; Gee et al., 2017; Cigna and Sowter, 2017; Alshammari et al., 2020; Alshammari et al., 2018; Bradley et al., 2021; De La Barreda-Bautista et al., 2022; Van Huissteden et al., 2021). Measures of surface subsidence of permafrost affected areas and the rate of ground motion can give an indication of permafrost degradation extents and rates. Furthermore, the ASPIS-InSAR data is unaffected by clouds, is based on freely available Sentinel-1 data, and can be used to analyse impact at continental scale. This means that ASPIS-InSAR can be used to identify areas that are at risk of increased $CH_4$ emissions and help to assess the radiative forcing ecosystem feedback resulting from permafrost thaw.

Complimentary remote sensing derived vegetation maps can be used to extrapolate plot based $CH_4$ emission data to the wider landscape (Varner et al., 2022). However, the scale-dependency of ecosystem properties that are non-linearly distributed in space (Siewert, 2018; Siewert and Olofsson, 2020) means that the spatial resolution of the remotely sensed data are important. Coarse spatial resolution remote sensing data, such as that from openly available and high temporal optical satellite-borne sensors, run the risk of missing small areas of those vegetation types with high $CH_4$ emissions, which may lead to an underestimate of total emissions. Further, the relatively slow rate of the response of vegetation to ground condition changes resulting from permafrost degradation means mapping vegetation may not fully explain patterns of $CH_4$ emissions. After subsidence starts to occur, the original vegetation type can persist for many years (>10 years) until the site conditions have changed to a point that the original vegetation is outcompeted (Johansson et al., 2013). This makes it difficult to use vegetation change in isolation as a year-on-year proxy for permafrost degradation and linked potential regulation of $CH_4$ emissions. Furthermore, permafrost degradation and linked increases in $CH_4$ production, is driven by several processes across scales. Variation in root inputs of labile carbon or oxygen in the peat profile occur over millimeters and can vary seasonally in response to water levels. Lateral erosion and subsidence linked to permafrost degradation occur on meter scales when measured over several years, while changes in the vegetation community from raised palsa vegetation to fen type vegetation occur over tens of meters at decadal time-scales. Together the different processes associated with permafrost degradation result in a spatially varied landscape with a mixture of functionally distinct geomorphological units with regard to soil moisture content and vegetation productivity, creating a palimpsest of processes and properties determining the effective greenhouse gas feedback of these regions (Siewert et al., 2021). Therefore, remote sensing methodologies which capture fine-scale spatial variation in vegetation composition, in addition to permafrost degradation over seasonal to yearly time scales, are needed to provide appropriate observations of degrading peatlands in the Arctic in order to measure their $CH_4$ emissions accurately.

With a view to estimating $CH_4$ emissions resulting from permafrost degradation in peatlands, this paper explores two remote sensing technologies to provide data for extrapolating plot-based methane measurements over sub-Arctic peatlands of northern Sweden. Specific objectives were to (i) carry out detailed field surveys to capture small-sized or local-scale variation in $CH_4$ emissions linked to vegetation type and permafrost subsidence; (ii) conduct vegetation mapping from Sentinel-2 satellite data, and in particular demonstrate the value of rigorous map validation on the quantification of $CH_4$



emissions; and (iii) quantify CH$_4$ emissions from areas experiencing permafrost degradation and those at high risk of subsidence as determined from Sentinel-1 InSAR (processed using ASPIS) data.


## 2 Methods

### 2.1 Field site description

Three peatland locations affected by permafrost conditions - the Tourist Station, Storflaket, and Stordalen wetlands (Figure
1) - situated to the south of Lake Torneträsk near Abisko, northern Sweden (68°12´N, 19°03´E, 351 masl), were the focus of study. The study sites are in sub-Arctic climatic zone, the mean annual temperature (MAT) ranges between 0.8 and 1.0 □C and the mean annual precipitation (MAP) ranges from 304 and 424 mm, in the West and in the East, respectively. The study sites comprise areas of raised palsas (i.e. relatively dry peat plateaus) and waterlogged areas with fen or birch forest vegetation. All three sites show signs of active permafrost thaw indicated by a lowering of the ground surface (i.e. ground
subsidence), high soil moisture levels, and areas of palsa collapse (i.e. complete permafrost degradation and transition from palsa to wetland) (Åkerman and Johansson, 2008).

Previous mapping of the vegetation at these sites using Unoccupied Aerial Vehicles (UAVs) and subsidence estimated using ASPIS-InSAR (De La Barreda-Bautista et al., 2022) show the vegetation on the palsa to be composed of bryophytes (e.g. *Sphagnum fuscum*), lichens and dwarf shrubs (*Empetrum nigrum, Andromeda polyfolia,* and *Betula nana*
(Sjögersten et al., 2016)). The most common herbaceous species was *Rubus chamaemorus*. Recently collapsed areas adjacent to and within the plateau areas tended to be vegetated by *Sphagnum* sp. and *Eriphorum vaginatum* and *E. angustifolium*. Extensively collapsed and subsequently flooded areas tended to be vegetated by graminoids, mainly *Carex* and *Eriophorum* species, herbaceous plants mainly *Menyanthes trifoliate, Comarum palustre*, and deciduous shrubs, such as *Salix laponica*, *Salix ssp.* and *Betula nana*. Forested wetland areas tended to be located adjacent to streams and at the
perimeters of the wetland areas. The mean peat depth was ca 60, 94, 50, and 40 cm, on palsas, *Sphagnum* sp., sedge, and forested wetland areas, respectively; mean soil C storage in the top 100 cm was 52, 34, 33, 40 kg SOC m$^{-2}$ for palsa, *Sphagnum* sp., sedge, and forested wetland, respectively (Siewert, 2018). The depth of the active layer (the seasonally thawed upper part of the soil) varies between drier areas, which have a shallower (ca 30 – 60 cm) active layer, and wetter areas, where the active layer is deeper or where there is no permafrost at all in the upper 1.5 m. In the Torneträsk area, the
active layer depths have increased, between 0.2 and 2.0 cm yr$^{-1}$ over 1978-2006, with a higher rate in more recent years (Åkerman and Johansson, 2008).



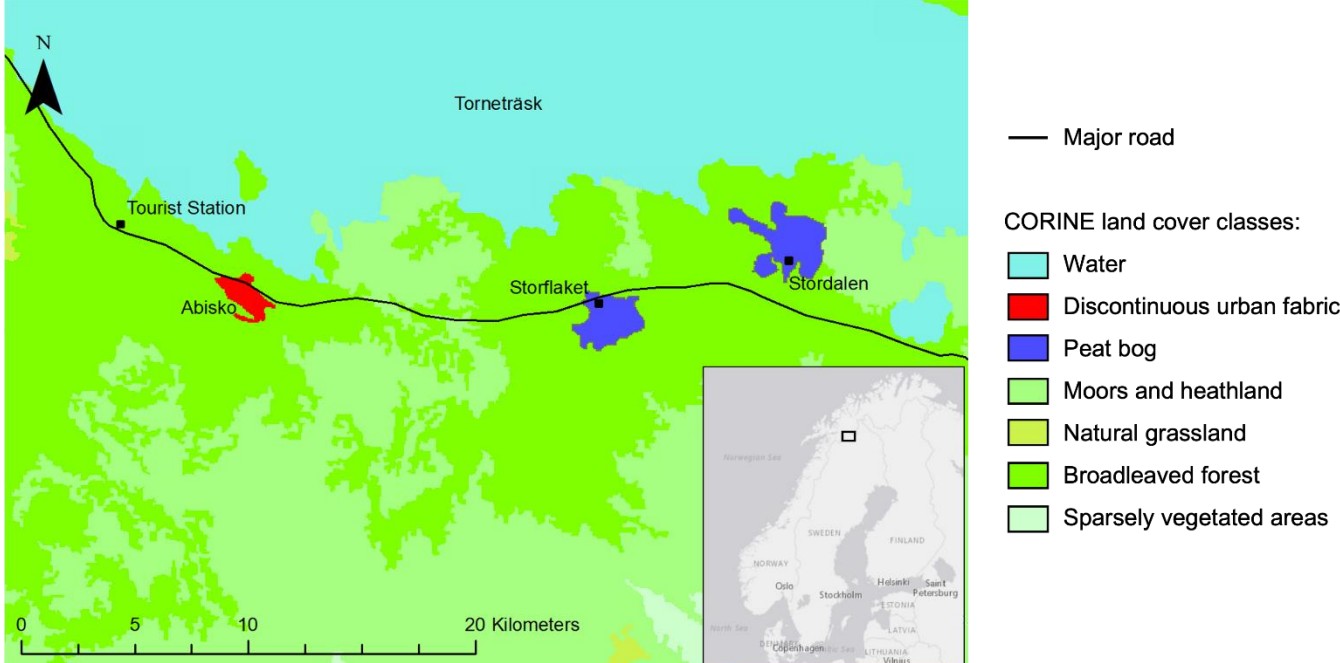

**Figure 1. Map showing the location of the three permafrost peatland study sites: Tourist Station, Storflaket and**
**Stordalen. Inset: Location of Abisko in northern Sweden. The landcover classes were based on the CORINE land**
**cover classification.**

## 2.2 Experimental design

The fieldwork to measure $CH_4$ emissions at the three study sites was carried out across the dates of 8-15 June, 18-29 of July
and 8-12 of September 2021. At each site, we targeted six common vegetation types, dry lichen, dwarf shrub, moist moss,
sphagnum wetland, sedge wetland and willow wetland. We also measured $CH_4$ emissions from subsiding areas with dry
lichen, dwarf shrub and moist moss vegetation types. Measurements in each vegetation type were carried out in blocks of ca
10 m diameter with 3-5 sampling points in each block. Sampling blocks were distributed across the sites and for the majority
of vegetation types data from at least two spatially separate (i.e. > 100 m apart) blocks were collected. As the dry lichen
subsiding vegetation type was not common, this vegetation was sampled less than the other classes. To facilitate the
collection of spatially distributed data from the common vegetation types, we selected sites with relatively easy access (e.g.
sampling of water logged areas were made in parts of the sites accessible by boardwalks). This approach allowed us to
collect $CH_4$ flux data from a large number of sampling points with the aim to generate scalable data from each vegetation
type. A total number of 502 measurements were collected across the three sites and the three dates, with all vegetation types
measured in several locations at each site. Specifically, on the raised palsa and subsiding palsa areas, the total number of
measurements point were 47 and 9 for dry lichen (DL) and dry lichen subsiding (DLS), respectively, 42 and 68 for dwarf
shrub (DS) and dwarf shrub subsiding (DSS), respectively, 50 and 56 moist moss (MM) and moist moss subsiding (MMS),





respectively. In the flooded fen vegetation areas, 102, 78 and 50 measurement points were in *Sphagnum* wetland, sedge wetland and willow wetland, respectively. Detailed description of the vegetation types are in de la Barreda Bautista et al.
(2022). Additional measurements of soil temperature and moisture using a Theta probe connected to a Theta meter (Delta-T devices, Cambridge, UK) were carried out in each point were methane emissions was measured. The active layer depth were determined using a graded metal pole in a subset of plots (ca 10 of each vegetation type in each of the three mires in June, July and Sept 2021 (De La Barreda-Bautista et al., 2022).

**2.3 CH$_4$ emissions measurements**

Methane flux measurements were conducted using a dynamic closed chamber connected to a Los Gatos ultraportable GHG analyser (San Jose, California, USA). Two types of chambers were used, one 15 and one 40 cm tall, the diameter was 40 cm. The chambers were equipped with two push-fit connectors that acted as an inlet and outlet allowing the gas to flow from the chamber to the analyser and then to flow back into the chamber. After the chamber was placed on the measuring point, it was
gently pushed on a 5 cm deep groove in the peat to create an airtight headspace between the ground surface and the chamber walls. To reduce problems with bubbles being released into the chamber the chamber was first placed on the ground and then lifted and allowed to vent. The chambers were then placed back on the ground and the sampling tubes connected. The soil CH$_4$ concentration (in ppm) increment inside the chamber was measured during 4 to 6 minutes, with data automatically recorded every 20 seconds. If the change in CH$_4$ concentrations was not linear, the measurement was stopped, chambers
lifted from the soil and a new measurement was taken after CH$_4$ readings had stabilized.

Trace gas concentrations, determined by the Los Gatos instrument, were converted to mass units using the ideal gas law (Eq. 1). Thereafter, using of the slope of the linear change in gas concentration over time and Eq. 2, CH$_4$ fluxes were calculated. Positive fluxes indicated GHG emissions from the soil into the atmosphere and negative fluxes indicated and uptake of atmospheric GHG by the soil.


$$n = \frac{PV}{RT} \tag{1}$$

$$F = \frac{Slope \times Volume_{chamber}}{Area_{chamber}} \tag{2}$$

where, $n$ is the number moles of trace gas (mol L$^{-1}$), $P$ is the atmospheric pressure (Pa), $V$ is volume of trace gas per litre of air (L L$^{-1}$), $R$ is the gas law constant (8314.46 L·Pa·K$^{-1}$·mol$^{-1}$), and $T$ is the temperature (K). $F$ is the calculated flux for either CH$_4$ or N$_2$O (mol m$^{-2}$ hr$^{-1}$), $Slope$ is the slope of the linear regression (mol L$^{-1}$ hr$^{-1}$), $Volume_{chamber}$ is chamber headspace volume (L) and $Area_{chamber}$ is the area of the chamber (m$^2$). Fluxes were converted to µg m$^{-2}$ h$^{-1}$ using the molecular weight of CH$_4$ (i.e. 16.04 g mol$^{-1}$, respectively). To avoid excluding very small CH$_4$ fluxes from the annual
calculations, fluxes with low r$^2$ were used in the calculation of cumulative emissions. By contrast, large CH$_4$ fluxes with low





$r^2$ were discarded as these fluxes were considered to be either affected by $CH_4$ ebullition, gas leakage or gas under pressured during transportation of vials. Seasonal fluxes were calculated to the thaw period (i.e. spring-autumn) by scaling the $CH_4$ flux measurements made in June, July and September to the length of the spring season, summer season and autumn periods, respectively, and then summing these. The length of the three seasons was determined using a combination of seasonal NDVI data from the Torneträsk valley station (Siewert and Olofsson, 2020) and long term air temperature data from Abisko (www.smhi.se).

## 2.4 UAV-captured data for Digital Elevation Models (DEMs) and vegetation mapping

Multispectral and true colour RGB data ultra high resolution (UHR) data were captured across Storflaket and Stordalen in 2020 and from the Tourist St. site in 2021 from ca 106 and 100 m height using a fixed-wing UAV - Sensefly Ebee fitted with a Parrot Sequoia multispectral sensor. The Parrot Sequoia obtained spectral measurements across four bands: Green (550nm ± 40nm), Red (660nm ± 40nm), Red Edge (735nm ± 10nm) and Near Infrared (790nm ± 40nm). The images were processed using Pix4D (Berlin, Germany) to produce an orthomosaic for each spectral band. This resulted in a ground resolution of ~11-13 cm for the multispectral and 2-3 cm for the RGB data. To ensure the orthomosaic and DEM were accurately geolocated we collected ca 50 ground control points using a differential GPS (dGPS;Trimble R8s) across all three sites. Further details on the UAV data collection are in de la Barreda Bautista et al. (2022).

The multispectral UAV data was resampled to 0.5m spatial resolution and then classified to produce a vegetation type map for the three study sites using the same nine classes (i.e. the dominant vegetation types found on raised palsa plateaux's, dry lichen, dwarf shrub and moist moss vegetation and areas covered by those vegetation types impacted by subsidence, plus three vegetation types which dominate the flooded areas: *Sphagnum* moss, sedge and willow wetland) as those for which *in situ* $CH_4$ flux data were measured. A Support Vector Machine classification approach (Foody and Mathur, 2004) was used in ArcMap 10.4 (ESRI) using the Train Support Vector Machine Classifier in the Spatial Analyst package, with simple random sampling of ground data in June 2021 for Storflaket ($n = 258$) and Stordalen ($n = 395$) sites and August 2021 for the Tourist Station ($n = 657$). The ground points from each site were randomly split into training and validation datasets of 70% and 30% proportions.

## 2.5 Sentinel-2 remotely sensed data for vegetation mapping

We used a neural network classification approach (Xie et al., 2008) within QGIS desktop 3.16.5. to predict vegetation cover over the three sites of interest using Sentinel-2 data (wavebands 2, 3, 4, 5, 6, 7, 8a, 11 and 12) and elevation, slope and roughness data derived from the ArcticDEM, optical-stereo imagery model at 2 m spatial resolution (Morin et al., 2016). To match the 20 m × 20 m resolution of the Sentinel-2 pixels, an average elevation was computed for each pixel. Sentinel-2 wavebands 2, 3 and 4 were re-sampled from 10 m to 20 m resolution using nearest neighbour interpolation to match the resolution of the other wavebands. The UAV-derived vegetation maps were used to train the Sentinel-2 scenes covering the three study sites of the Tourist Station, Storflaket and Stordalen - for this the UAV data was upscaled to 20 m × 20 m using





majority classification, this yielded a dataset of 2376 pixels. Training and test datasets were created from this. The testing set used was generated as a random stratified sample comprising 50 pixels per vegetation class, unless there was less than that (e.g., the willow wetland class had only 20 test pixels).

The neural network approach was used to classify the Sentinel-2 data into vegetation classes from which the total areal extents of each vegetation class across the three sites were calculated. Specifically, a multi-layer perceptron network

with backpropagation and weight backtracking was used, therefore the 'learning rate' and 'momentum' parameters were not required (Xie et al., 2008). Two neural networks were generated: one that produced a standard hard classification (i.e., a single class label is produced for each pixel) and one that produced a soft classification (i.e., grades of class membership is produced for each pixel). The optimal model structure for the majority and the soft classification approaches consisted of twelve principal components, normalised between 0-1, one hidden layer with three neurons (hard classification model) or ten

neurons (soft classification model) and an output layer of fifteen units corresponding to the land cover types identified within the three study sites. Both models were developed through trial and error and iteration to arrive at the maximum overall accuracy. The soft classification was undertaken as it enables the estimation of the proportion of different vegetation types within each pixel offering greater understanding of the area covered by each class, in particular to capture and map those smaller patches of vegetation that have a larger contribution to $CH_4$ flux.

Misclassification errors in the maps may substantially bias estimates of class areal extent generated from the maps. An adjustment for misclassification bias in a map can be made if its accuracy has been rigorously evaluated. Here, good practice methods for accuracy and area estimation (Olofsson et al., 2013; Olofsson et al., 2014) were followed to produce misclassification bias adjusted estimated of class area; an adjustment for the soft classification would ideally require more detailed ground reference data than was available. Thus in total,  three vegetation type outputs (maps and class areal extents)

from the Sentinel-2 data were produced (soft classification; hard classification and misclassification bias hard classification). The areal estimates of the vegetation classes were subsequently multiplied with the mean $CH_4$ fluxes measured for each vegetation class to produce $CH_4$ budgets for the entire mapped area.

**2.6 Surface motion from Sentinel-1 InSAR**

We used C-band SAR data from the Sentinel-1 satellites (European Union's Copernicus Programme) to map multi-annual average velocity, subsequently denoted surface motion of the study areas. Details of this approach are in de la Barreda Bautista et al, 2022. Briefly we gathered images from 2017 to 2020, focusing on the thaw season between April and October for three separate years. A total of 125 images were acquired from the descending mode and surface motion was obtained

using the APSIS method (formerly known as ISBAS), which relaxes the requirement for consistent phase stability for all observations using a coherence threshold of 0.45. Following InSAR processing the ground resolution was 20 m x 20 m. Using a stable reference point located in the centre of Abisko village first the multi-annual average velocity (or average surface displacement) between 2017 and 2020 (with negative values representing subsidence) was calculated and second, for





each season, a time series of surface motion was produced, showing the change in relative height of the surface. Surface
motion range for the 2017 and 2020 thaw periods were calculated by subtracting the minimum surface motion value from the
maximum surface motion value within respective time series between June and October. The InSAR data were thereafter
compared to the $CH_4$ flux data as described in the statistical analysis section below.

**2.7 Statistical analysis**

We used residual maximum likelihood (REML) mixed models to test for significant differences in $CH_4$ fluxes among sites
and vegetation and their interactions. For this analysis we used vegetation type as the fixed effect and sampling location (i.e.
block), month and site as random effects. The residuals were inspected to ensure the normality assumption of the analysis
was met. The standard error of the difference was used to evaluate which vegetation types that differed from each other. We
tested for the relationships between the UAV predicted $CH_4$ budgets per vegetation type and the three different Sentinel-2
based classifications using linear regression analysis. Relationships between ASPIS-InSAR linear motion between the 2017
and 2020 thaw periods and seasonal $CH_4$ fluxes were explored using REML in which the ground motion categorised into 2.5
mm ranges were used as the fixed effects and site as the random effect. For this analysis comparison, the $CH_4$ fluxes
extrapolated for each target vegetation type to the three sites using the VHR UAV data, which are able to capture the local-
scale variation in the vegetation types, were aggregated to the 20×20 m pixel size of the InSAR data.


**3 Results**

**3.1 Variation in $CH_4$ fluxes linked to vegetation type and permafrost degradation**

The intact raised palsa vegetation types (DL, DS and MM) had generally low $CH_4$ fluxes; the dry lichen vegetation type even
acted as a weak $CH_4$ sink in spring (Figure 2a). The areas identified as showing signs of permafrost degradation/subsidence
had higher soil moisture content ($F_{8, 489} = 167.91$, P <0.001; SED = 3.6%; Figure 3a) and deeper active layer depths ($F_{7,163=}27$
P < 0.001 and $F_{8,91} = 8.15$, P < 0.001, for July and September, respectively; Figure 4) than areas that were less affected by
degradation. Degrading areas had elevated $CH_4$ fluxes compared to the corresponding vegetation types in areas that did not
display visual signs of degradation ($F_{8, 489} = 8.15$, P <0.001; SED = 2.85 mg $CH_4$ m$^{-2}$ h$^{-1}$), with the largest increase in
emissions being for subsiding areas of dwarf shrub (Fig 2). The emission from the subsiding palsa areas was highest in
summer corresponding to the period of highest air and soil temperatures (Fig 2 b, Fig 3). The highest emissions were from
the flooded fen vegetation types, where emissions peaked in autumn (Fig 2 c). Temperature peaked in July at the non-
flooded sites while at flooded sites temperatures remained relatively constant after the initial thaw period. The coldest
temperature were measured in subsiding DL and DS areas in June ($F_{8, 490} = 9.06$, P < 0.001; SED = 0.7□C).





**Figure 2. CH₄ fluxes measured in different vegetation types in a) June, b) July, c) September and d) cumulative seasonal CH₄ emissions, "sub." denotes measurements made in subsiding areas of vegetation types found on the**
**raised palsa plateaus. For September, no data are available from the dry lichen sub. class. Means and SE are shown.**

Integrated CH₄ fluxes per vegetation type showed highest seasonal fluxes of ca 400 kg CH₄ ha⁻¹ season⁻¹ from willow wetland areas with emissions from *Sphagnum* and sedge wetland vegetation types being ca half those of the willow




areas at ca 200 kg $CH_4$ $ha^{-1}$ $season^{-1}$. The seasonal emissions from the subsiding dwarf shrub, moist moss, and dry lichen
areas were 50 and 25 kg of $CH_4$ $ha^{-1}$ $season^{-1}$, respectively, which were higher than in un-degraded areas where emission
from raised palsa vegetation types were low at less and/or equal to 2 kg of $CH_4$ $ha^{-1}$ $season^{-1}$.

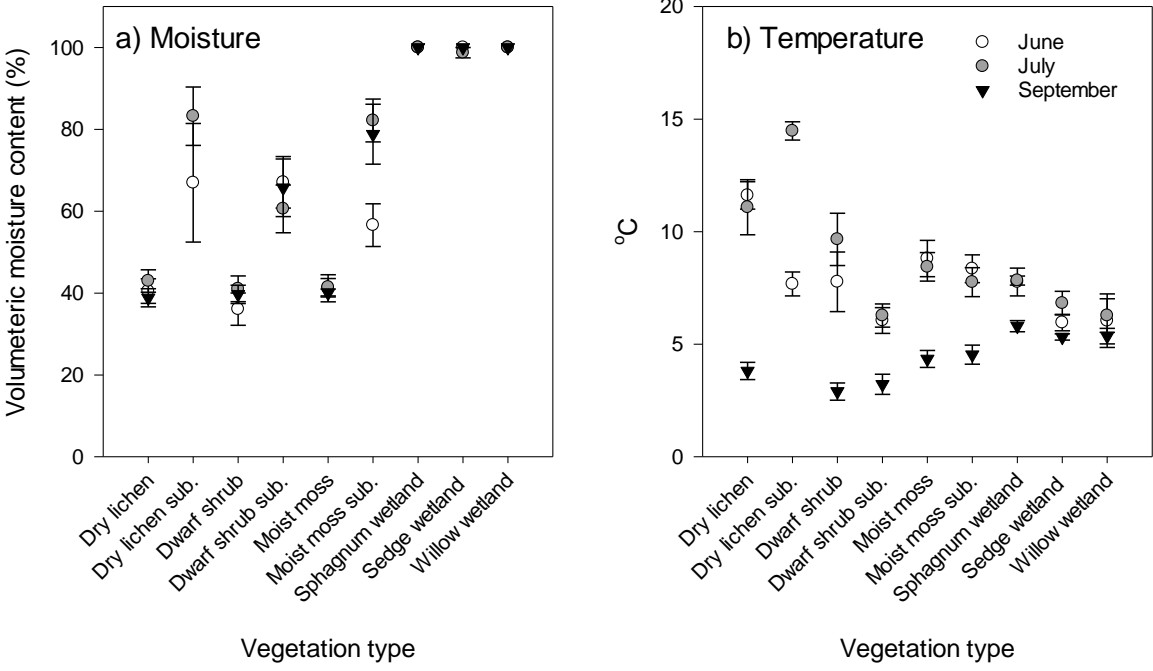

**Figure 3. Variation in a) moisture and b) temperature among vegetation types measured in June, July and September, "sub." denotes measurements made in subsiding areas of vegetation types found on the raised palsa**
**plateaus. For September no data is available from the dry lichen sub. class. Means and SE are shown.**



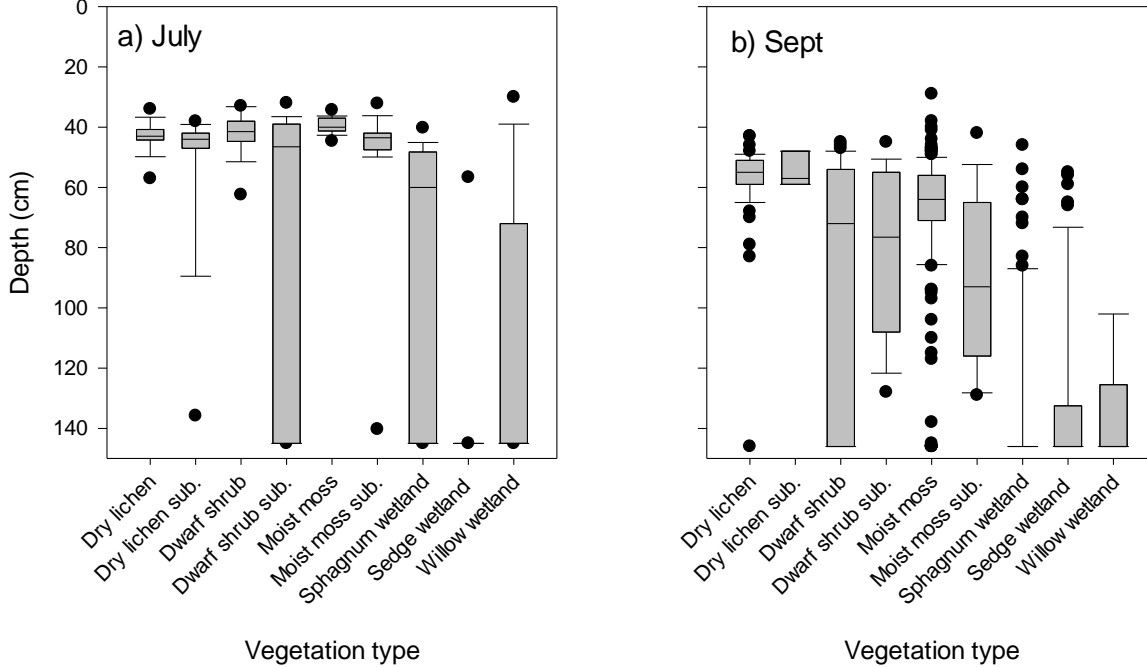

**Figure 4. Variation in active layer depth among vegetation types in a) July and b) September, "sub." denotes measurements made in subsiding areas of vegetation types found on the raised palsa plateaus.**

**3.2 Scaling CH₄ fluxes via vegetation mapping and calculation of class areal extent**

The mapping of the nine vegetation classes found on degrading palsa mires using the VHR UAV data gave an overall classification accuracy of 71% (confusion matrices were generated in base R; Supplementary information 1). The area covered by the target wetland vegetation classes (i.e. excluding rock outcrops, birch forest areas, water bodies and anthropogenic surfaces) across the three sites was ca. 50 ha while the total surveyed are was 83.7 ha (Table 1). Subsequently % estimates of areas are made relative to the target vegetation area. Areas where the visual inspection of the surface topography in the field clearly indicated advanced subsidence equated to 10.0 % of the area covered by the vegetation types. The three subsiding vegetation classes covered 1.5, 2.4 and 1.1 ha, or 3.0, 4.8 and 2.2 % of the target vegetation type area, for dry lichen subsiding, dwarf shrub subsiding and moist moss subsiding, respectively (Table 1). Key here was that the classification was of a high enough accuracy for this output to be used as the "ground" data for the Sentinel-2 classifications.

**Table 1. Area of different wetland vegetation types at each of the three study sites and their proportional (Prop.) contribution to the area covered by the target vegetation types. (Please note that area of anthropogenic, exposed rock,**





**sparse birch forest on mineral soil, forested wetland and water as well as pixels impacted by shadows are included in**

**the CH₄ flux study and hence they are not shown in the table).**

| Vegetation | Tourist St. | | Storflaket | | Stordalen | | |
|---|---|---|---|---|---|---|---|
| | Area (ha) | Prop (%) | Area (ha) | Prop (%) | Area (ha) | Prop (%) | ☐ Sites (ha) |
| Dry lichen | 0.002 | 0.1 | 0.177 | 1.0 | 0.245 | 0.8 | 0.4 |
| Dry lichen sub | 0.004 | 0.1 | 1.483 | 8.3 | 0.012 | 0.0 | 1.5 |
| Dwarf shrub | 0.494 | 15.0 | 0.897 | 5.0 | 1.785 | 6.1 | 3.2 |
| Dwarf shrub sub | 0.681 | 20.6 | 0.450 | 2.5 | 1.291 | 4.4 | 2.4 |
| Moist moss | 0.144 | 4.4 | 4.861 | 27.3 | 5.612 | 19.3 | 10.6 |
| Moist moss sub | 0.403 | 12.2 | 0.143 | 0.8 | 0.571 | 2.0 | 1.1 |
| *Sphagnum* | 0.827 | 25.1 | 2.250 | 12.7 | 7.072 | 24.3 | 10.1 |
| Sedge | 0.351 | 10.6 | 4.835 | 27.2 | 5.987 | 20.6 | 11.2 |
| Willow | 0.392 | 11.9 | 2.681 | 15.1 | 6.523 | 22.4 | 9.6 |
| ∑Target vegetation | 3.3 | | 17.8 | | 29.1 | | 50.2 |

The resultant classification vegetation maps from the Sentinel-2 hard and soft classifications had overall accuracies of 50.2%

and 83.2%, respectively. The class areal extents from these classification outputs, along with class areal extents obtained

when hard classification output was adjusted for misclassification bias, are presented in Table 2 (which depicts those classes

shown in Table 1). The hard classification of Sentinel-2 data did not capture the fine-scale vegetation mosaic on the raised

part of the palsa mire (i.e., dry lichen, and dwarf shrub prior to subsidence), instead it mapped the vegetation types in those

areas as either moist moss or dwarf shrub subsiding vegetation classes, leading to the lower overall classification accuracy

than obtained using the soft classification approach. Once corrected for misclassification bias, the hard classification areal

extents for each vegetation class improved considerably in that the estimates are comparable to that mapped using the UAV

data and the soft classification of the Sentinel-2 data. The soft classification of the Sentinel-2 data was able to map all nine

target vegetation types and followed the same distribution of more or less abundant vegetation types, predicting similar total

class areal extents as that of the UAV-derived maps. In both cases (bias corrected hard classification and soft classification),

those vegetation types on the raised palsa area missed by the hard classification, have a calculated areal extent.

The soft classification of the Sentinel-2 data produced a higher estimate of the areal extent of the fen vegetation

types at 35.4 ha than the UAV classification but succeeded in mapping all the vegetation classes present across the study site.

The mapping from the hard classification of Sentinel-2 under-estimated the fen vegetation types slightly at 28.64 and missed

some of the other vegetation classes (i.e., Dry lichen, Dry lichen subsiding and Dwarf shrub). After correcting for





classification bias, the areal extents of vegetation classes from the hard classification closely matched to those of the UAV mapped areas, with 32.21 ha of the fen vegetation types estimated to occupy the study site.

Based on the UAV-classification vegetation mapping (class areal extents), 30.9 ha (or 61%) of the area covered by the target vegetation types was comprised of the three vegetation classes (*Sphagnum*, sedge and willow wetland – known as fen vegetation types) associated with the highest CH$_4$ emissions. Using the mapping of areal extents from the UAV data to
model CH$_4$ fluxes shows marked differences in CH$_4$ fluxes between raised palsas and the wetland types surrounding these (Figure 5; Table 3). Areas of the raised palsa, which are not showing visual signs of subsidence, generally have very low emissions while areas with degrading permafrost and subsidence, as well as those areas with no permafrost emit methane. Among the vegetation types that are associated with high CH$_4$ emissions, the maps highlight the importance of the willow wetland vegetation type as a source of methane from these wetlands.  This is because it have both high emissions per area
and covers substantial areas in particular at the Stordalen mire site. Although the highest CH$_4$ emissions are from the areas which have lost their permafrost that are surrounding the remaining palsa, there are indications of increased emissions also within the palsas.

**Table 2. Comparing class areal estimates for the target vegetation types using Sentinel-2 (S-2) hard and soft**
**classification and bias-corrected hard classification. For comparison the estimates are provided for the UAV-derived classification which served as the ground data.**

| | S-2 Hard Classification | S-2 Soft Classification | Bias-corrected S-2 Hard Classification | Ground data (UAV Classification) |
|---|---|---|---|---|
| **Vegetation type** | Area (ha) | Area (ha) | Area (ha) | Area (ha) |
| Dry lichen | 0.00 | 0.4 | 0.0 | 0.4 |
| Dry lichen sub | 0.00 | 1.2 | 0.8 | 1.5 |
| Dwarf shrub | 0.00 | 3.3 | 1.1 | 3.2 |
| Dwarf shrub sub | 2.9 | 2.1 | 5.3 | 2.4 |
| Moist moss | 13.3 | 8.1 | 13.7 | 10.6 |
| Moist moss sub | 0.0 | 1.3 | 0.3 | 1.1 |
| *Sphagnum* | 7.4 | 10.1 | 11.2 | 10.1 |
| Sedge | 19.4 | 13.2 | 11.7 | 11.2 |
| Willow | 1.8 | 12.1 | 9.3 | 9.6 |
| **∑ Target vegetation** | **44.8** | **51.0** | **52.8** | **50.2** |
| **∑ Fen with high CH$_4$ emission** | **28.6** | **35.4** | **32.2** | **30.9** |





Methane emissions from within the palsa areas are commonly associated with subsiding moist moss and subsiding dwarf shrub vegetation, as well as development of Sphagnum wetland vegetation in subsiding areas that have become waterlogged

either at the edges of the palsas or in their interior following degradation. The subsiding dwarf shrub vegetation is mainly found along the edges of the palsas that are experiencing lateral erosion while the subsiding moist moss vegetation type occurs in the interior parts of the palsa in areas with deeper active layers and the development of a hummocky surface.

When $CH_4$ emissions are integrated across the three wetland areas using the UAV-derived vegetation mapping net emissions are 8519 kg $CH_4$ season$^{-1}$ (Table 3). The total emissions are composed of 16.6, 142.6 and 8359.5 kg $CH_4$ season$^{-1}$

from raised palsa vegetation (DL, DS, MM), subsiding palsa vegetation (DLS, DSS, MMS) and waterlogged vegetation types (SW, SE WW), respectively. This shows that currently emissions from either intact areas in the early stages of degradation is low compared to the already degraded areas. However, as the areas of raised palsa vegetation and subsiding palsa vegetation are large (ca 20 ha (or 40% of the area covered by target vegetation types) when combined), there are substantial risks of higher $CH_4$ emissions from these sites associated with continued degradation of the currently subsiding

areas and onset of degradation of the remaining palsa areas.









**Figure 5. (a-c) Detailed vegetation maps separating out wetland vegetation types including vegetation types associated with raised and subsiding palsa areas at the Tourist st., Storflaket and Stordalen mires. The vegetation maps were**

**generated from UAV-captured data. (d-f). Maps of seasonal CH4 emission based on the seasonal CH4 emission measured in the nine target vegetation types across the three mire sites. The focus palsa-fen wetland area outlined in purple is the area were ground data on CH4 fluxes were collected as well as for the vegetation mapping. The full wetland area at each site is indicated by the black boundary line.**

**Table 3. Seasonal methane emissions for 2021 for the total area of each target vegetation classes within the area covered by the UAV flights. Methane emissions are estimated using three different models using Sentinel-2 data and one model using UAV data. Mean and SE are shown.**

| Vegetation | CH4 flux S-2 hard classification (kg CH4 season$^{-1}$) | | CH4 flux S-2 soft classification (kg CH4 season$^{-1}$) | | CH4 flux S-2 bias corrected hard classification (kg CH4 season$^{-1}$) | |
|---|---|---|---|---|---|---|
| Dry lichen | 0 | 0 | 0.05 | 0.11 | 0 | 0 |
| Dry lichen sub | 0 | 0 | 7.2 | 3.6 | 4.7 | 2.3 |
| Dwarf shrub | 0 | 0 | 7.0 | 5.5 | 2.3 | 1.8 |
| Dwarf shrub sub | 134.5 | 41.4 | 96.7 | 29.7 | 248.6 | 76.4 |
| Moist moss | 12.3 | 3.7 | 7.5 | 2.2 | 12.1 | 3.6 |
| Moist moss sub | 0.0 | 0.0 | 23.4 | 4.2 | 4.9 | 0.9 |
| *Sphagnum* wetland | 1422.0 | 400.3 | 1942.8 | 547.0 | 2159.4 | 607.9 |
| Sedge wetland | 4415.3 | 936.3 | 2993.5 | 634.8 | 2651.8 | 562.3 |
| Willow wetland | 726.2 | 184.7 | 4885.9 | 1242.8 | 3750.4 | 953.9 |
| ∑ **Vegetation types** | **6710** | | **9964** | | **8834** | |

Taking the Sentinel-2 sets of estimates of areal extent of vegetation types to estimate the CH4 emissions of the vegetation
types showed both over- and under-estimation of emissions (compared to the UAV-derived CH4 estimates) - the hard classification under-estimated CH4 emissions across the site by 21 %; while the soft classification over-estimated emissions by 17%. The mismatch between the Sentinel-2 hard classification and the UAV estimates were linked to a misclassification of willow wetland as sedge wetland (Table 2, Supplementary information 1). Correcting the output of the hard classification for misclassification bias improved the match of CH4 estimates to that derived from the UAV data, and key was that target
vegetation classes were included in the mapping following this post-classification process. For the soft classification, it was the misclassification of sedge and willow wetland area. Correlation analysis between the UAV- and Sentinel-2 classification outputs showed that the bias corrected hard classification was more closely correlated to the UAV predictions, followed



closely by the soft classification with $R^2 = 0.997$ and $R^2 = 0.995$ (P < 0.001 and P < 0.001), respectively. The hard classification performed least well with an $R^2 = 0.37$ (P < 0.001; Fig 6).

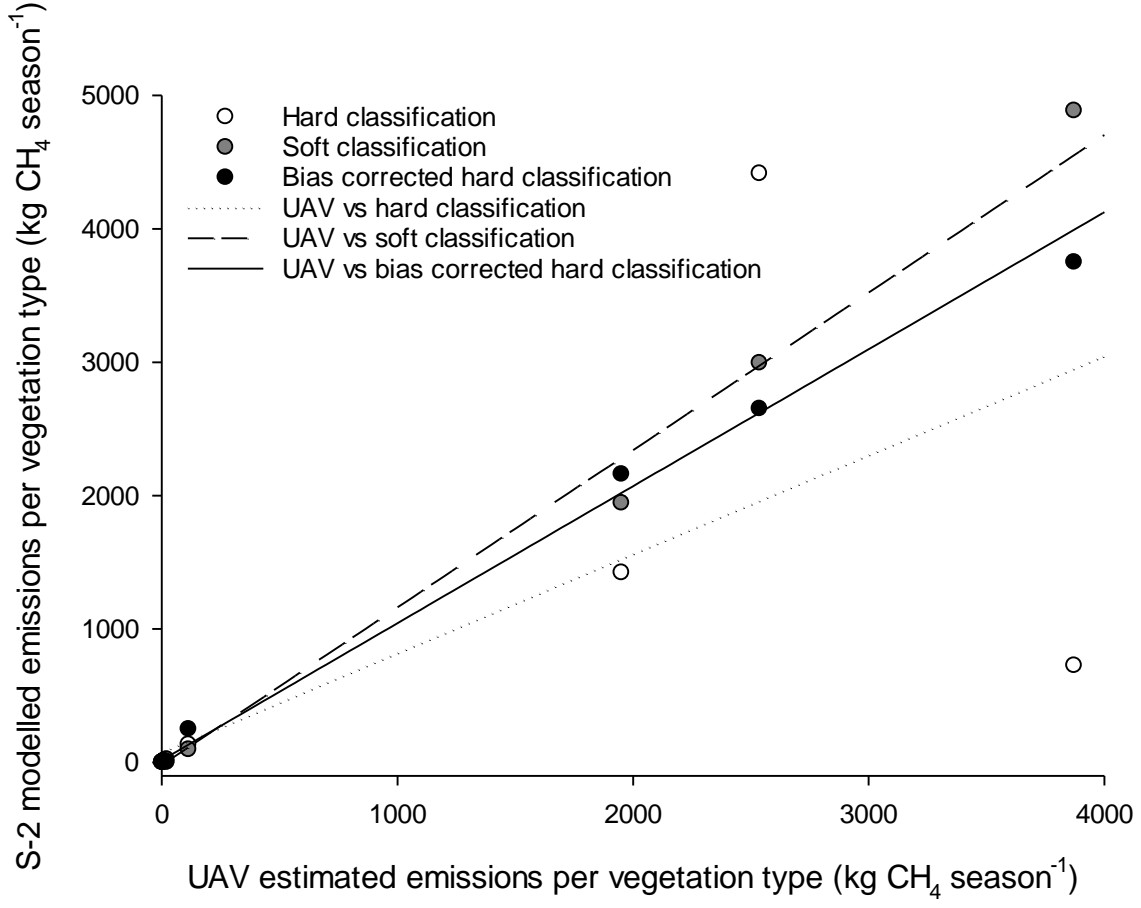

**Figure 6. Relationship between UAV estimated methane emissions per vegetation types and the three Sentinel-2 (S-2) models.**

### 3.3 Linking InSAR determined ground motion to CH$_4$ emissions

InSAR detected subsidence across 90 % of the study area, indicating that the majority of the surface of these sites were impacted by mm level rates of subsidence. However, rates of ground motion varied considerably across the study area and among vegetation types. Importantly, across the three sites, areas with relatively low levels of ground motion were associated with the highest mean CH$_4$ emissions ($F_{8,1135} = 5.63$, p < 0.001, SED = 961.5; Fig 6) while both uplifting and the most subsiding areas tended to have lower fluxes. High cumulative CH$_4$ emissions (summed over all pixels for a given range of ground motions) were found for areas that were either stable or subsiding at an intermediate rate (Fig 7) while the most subsiding areas had low cumulative CH$_4$ emissions.

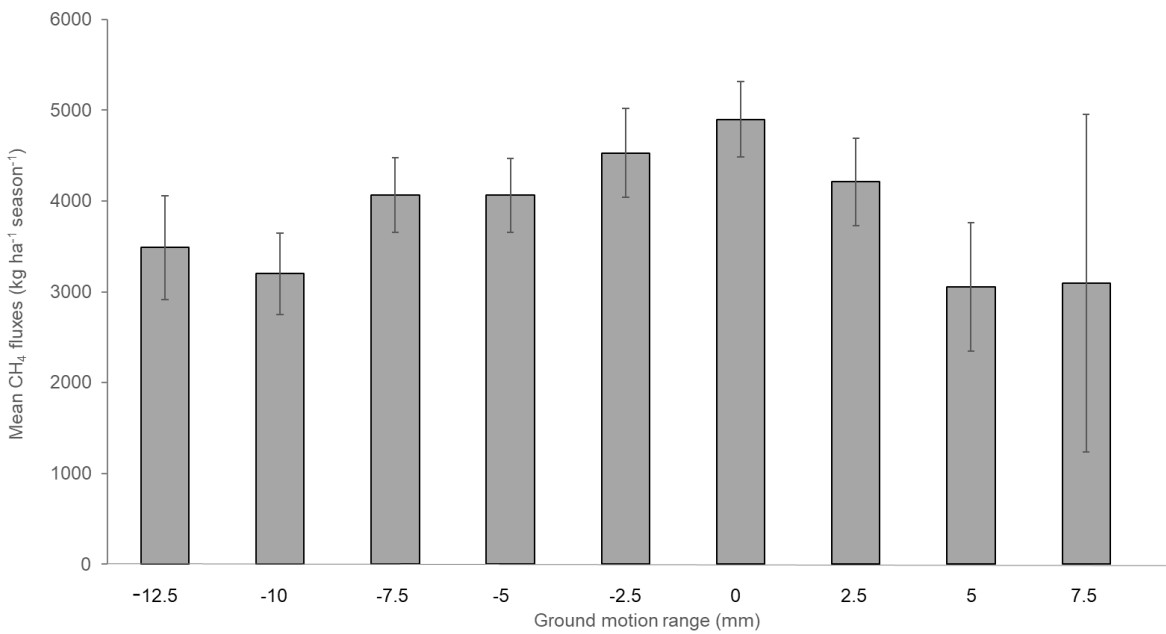

**Figure 7. Average seasonal CH₄ emissions for different ground motion ranges (categories in 2.5 mm ranges) across**
**the three study sites. Negative values indicate subsidence and positive values indicate uplift. Mean and SE are shown.**

This pattern of areas of high subsidence being linked to areas of low CH$_4$ emissions was also apparent in the subsidence and
CH$_4$ emissions maps (Fig 9). For example, at the Tourist St. mire areas of high subsidence (comprising mainly subsiding
moist moss vegetation Fig 5 a) corresponded to areas with relatively low CH$_4$ emissions while areas that have already
transitioned to *Sphagnum* or willow wetland are associated with bands of high CH$_4$ emissions along the degrading edge of
the palsa plateaux (Fig 5 a and d). Similar patterns were evident on the Storflaket mire, where degrading areas within the
palsa plateaux did not display high CH$_4$ emissions while areas that have transitions to *Sphagnum* or sedge wetland have high
CH$_4$ emissions (Fig 7 b and e). The Stordalen site showed the largest areas of high subsidence that traversed the boundary
between the palsa plateau and the fen areas (Fig 7 c and f). In this part of Stordalen the *Sphagnum* wetland vegetation type
was present both in the hollows in the degrading interior of the palsa plateaux and in adjacent areas where the palsas had
already collapsed, explaining the high CH$_4$ emissions (together with the presence of the willow wetland vegetation type)
from this area.





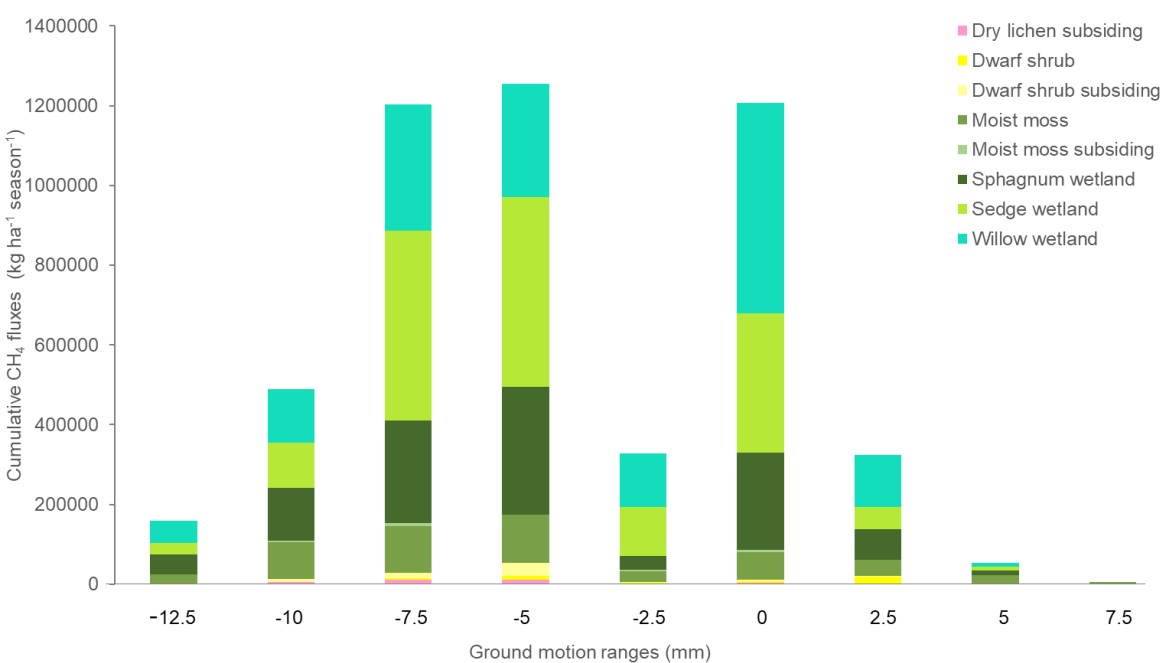


**Figure 8. Cumulative CH₄ emission (± SE) for different ground motions across the thaw seasons of 2017 and 2020 for pixels (i.e. a sum of the CH₄ emissions from all pixel of a given ground motion (categories in 2.5 mm ranges)) of the target vegetation classes. The dry lichen vegetation types did not cover large enough areas to be evaluated on the 20×20 m pixel scale of the ground motion data and is therefore not included in this comparison. Negative values**
**indicate subsidence and positive values indicate uplift.**

The net emissions of $CH_4$ across subsiding parts of the three wetlands (as detected by InSAR) were modest at 116 kg $CH_4$ season$^{-1}$. However, when these areas have degraded to such a point that they are fully waterlogged and have transitioned to either *Sphagnum* or sedge vegetation we estimate emissions from these, currently low emitting, areas to be around 6480 kg
season$^{-1}$ or if we use the highest emitting vegetation type as a predictor of future emissions we arrive at 12 960 kg season$^{-1}$.







**Figure 9. Maps of (a-c) ground motion (de la Bautista Barred et al., 2022) and (d-f) CH₄ fluxes at the Toursist st., Storflaket, and Stordalen mire. The black boxes show the areas referred to in the text.**


**4 Discussion**

Methane emissions varied strongly across the palsa peatlands with consistent differences among vegetation types over space and time. The low emissions from the intact dry palsa areas and trends of increased emission following degradation and

flooding were comparable with those reported from other part of Scandinavia and Russia (Nykänen et al., 2003; Olefeldt et al., 2013; Liebner et al., 2015; Miglovets et al., 2021; Varner et al., 2022; Glagolev et al., 2011). However, there was strong variation in the magnitude of that response depending on the degradation stage, level of flooding and vegetation community present. For example, as long as sites were mainly impacted by increased waterlogging but no vegetation change (i.e. the raised palsa vegetation persisted despite experiencing subsidence), CH₄ emissions only increased modestly (Fig 2). This

suggest that waterlogging *per se* was not sufficient to drive high emissions as these were only recorded following flooding and a transition of the vegetation community to *Sphagnum* mosses, sedges and other wetland adapted species. This is consistent with the literature were different type of sedge vegetation communities and thermokarst activity were associated with the higher end of emissions (Glagolev et al., 2011; Dzyuban, 2002; Walter et al., 2006). An important finding here is the high CH₄ emissions reported from fen and in particular the willow vegetation type. These emissions were consistent

across the three study sites and were at the upper range of emissions reported in the literature (Olefeldt et al., 2013; Miglovets et al., 2021) and in the range of emissions from ponds and thermokarst lakes in degrading permafrost in the literature (Walter et al., 2006; Glagolev et al., 2011; Dzyuban, 2002). Possible explanations for this variation among vegetation types and very high CH₄ emissions are the impact of the vegetation on CH₄ emission both via plant mediated transport, root exudate quality and quantity and the level of rhizosphere oxidation. For example, both *M. trifoliate* and in

particular *C. rupestris* are known to transport CH₄ through their vascular tissues while other species common in our system e.g. *S. lapponica* do not share this trait (Ge, 2022). An additional way in which plants may contribute to high CH₄ production and emissions are by releasing labile carbon in the form of root exudates into the rhizosphere. This carbon flow can represent a substantial part of NPP and can have a large impact on CH₄ emissions (Ström et al., 2003; Girkin, 2018). We speculate that some of the willow vegetation type species, *S. lapponica*, and other *Salix* ssp, and *Comarum palustre*, which were abundant

within the willow wetland vegetation type, may drive high emissions via root exudation or those species may reflect specific edaphic conditions that are favourable for CH₄ production. Further, root exudation may be an explanation of high emissions also from the tall sedge vegetation type. However, further studies would be required to establish if the high emissions from the willow wetland vegetation community in our study is consistent across greater spatial ranges as well as the specific drivers of the high emissions. Given the variation in flux rates among areas with different vegetation types, despite broadly

similar moisture conditions, it is also likely that the species composition within the wide vegetation classes often used for describing permafrost peatlands (raised palsa and fen) directly impact the magnitude of the CH₄ emissions as these peatlands



degrade. Hence improved understanding the plant species specific mechanisms that control $CH_4$ emissions beyond abiotic drivers (such as water logging, redox, soil and air temperature) (Olefeldt et al., 2013) are important for estimations of landscape level $CH_4$ emission and how these will be altered as these palsa peatlands continue to degrade and their vegetation
composition change.

The fact that small extents and local-scale variation in ground/vegetation conditions strongly impact net emissions is important when it comes to developing spatial models for predicting $CH_4$ emissions over large areas. Our three models using Sentinel-2 data differed in their ability to detect the local-scale variation in the vegetation. The hard classification model did not succeed in capturing this variability and underestimated the total wetland area. The soft and adjusted hard
classification was both successful in estimating the overall area of the target vegetation types compared to the UAV estimated areas. When the areal estimates from the three models were used for extrapolating the $CH_4$ emissions only the adjusted hard classification estimated the emissions within a 5% error margin to the UAV estimated fluxes. The hard classification underestimated fluxes by 21 % while the soft classification over estimated emissions with 17%. This means that Sentinel-2 (or other freely available satellite data products e.g. Landsat) have sufficient spatial resolution to be an
effective tools for scaling up $CH_4$ emissions linked to particular vegetation types in the local scale vegetation mosaic that composes degrading peatland. However, among the three Sentinel-2 models we used, the adjusted hard classification clearly outperformed the other two models. Both, the unadjusted hard or soft classification, would have included considerable error when used to extrapolate the emissions to regional scale due to scale-dependency leading to inaccuracy in the relative distribution of individual land cover classes at coarser resolutions (Siewert, 2018; Hugelius et al., 2020). This highlights both
the risks of substantially increasing the error of the modelled estimates and the opportunity offered by the adjusted hard classification approach for scaling of $CH_4$ emissions using remotes sensing data (Olofsson et al., 2013; Olofsson et al., 2014).

The implication of this is that current estimates of changes in GHG emissions following permafrost degradation that often only use the distinction between palsa and fen (e.g. (Nykänen et al., 2003; Olefeldt et al., 2013; Miglovets et al., 2021;
Varner et al., 2022) may result in considerable uncertainty and substantial underestimation of $CH_4$ emissions linked to degradation. This is because although this simple classification will reflect the major shift from dry to wet conditions and the large increase in $CH_4$ emissions that are associated with this, hot spots of high emissions will be missed as well as variation in $CH_4$ emissions among different fen vegetation types. The role of different vegetation types is evident from our $CH_4$ data from a range of wetland vegetation communities found at the study sites. These shows that variation in $CH_4$ emissions
among common vegetation types within the "fen" vegetation class can result in nearly twice as high emissions on an areal basis. Here the detailed vegetation mapping capabilities of drones provide an important way forward together with higher resolution aerial photography and satellite data e.g. WorldView, Quickbird, and Planet Ldt's "Super Dove mission when upscaling plot data to the landscape scale (Siewert and Olofsson, 2020; Siewert et al., 2015).

The relatively low $CH_4$ emissions from areas with high subsidence (Fig 7-9) reflect both the fact that they (i)
represent the permafrost degradation front, which is spatially less extensive than already degraded or stable areas, and (ii)



that although the subsiding areas of a given vegetation type (Fig 2) release more $CH_4$ than their intact counter parts, emissions are low compared to areas which are fully degraded and flooded. Further, although $CH_4$ emissions increased as a specific vegetation types showed signs of permafrost degradation and subsidence (Fig 2) the initial increase in $CH_4$ emissions at the most subsiding sites is modest compared to the high $CH_4$ emissions in areas where degradation has

progressed to a point were permafrost has thawed all together and subsidence has slowed down (albeit some consolidation of the peat material is still ongoing as suggested by subsidence rates at a lower rate in areas which have lost their permafrost).

Our work suggest that subsidence measured using ASPIS-InSAR can be used to predict increases in $CH_4$ emissions from areas experiencing permafrost degradation. The InSAR method also offers a highly sensitive early warning system of the initial stages of degradation. This is valuable because early stages of permafrost degradation are difficult to capture using

optical data or ground surveys, which tend to rely on changes in the vegetation or increased surface heterogeneity which only become apparent after the degradation has progressed substantially. This is evident in the different areal estimates of the subsiding area using the two different vegetation mapping approaches using UAV and Sentinel 2 data. Mapping for subsiding areas using ground truthing of VHR UAV optical data estimated the subsiding area of the study sites to ca 10% while the InSAR data detected subsidence in 90% of the wetland area. The implication of the InSAR data is that the majority

of the palsa peatlands area at our study sites is already experiencing permafrost degradation. Indeed, ongoing large-scale permafrost degradation at the sites is further supported by analysis of historical orthophotos and long term satellite data analysis as well as model predictions that suggest that permafrost in wetland will no longer be supported in northern Scandinavia by 2050 (De La Barreda-Bautista et al., 2022; Varner et al., 2022; Fewster et al., 2022). This raises concerns for the fate of the ca 2000 ha of palsa peatlands in Sweden over the next decades (Backe, 2014). The evidence of large-scale

degradation at our study sites and the potential for very high $CH_4$ emissions from areas that currently have low emissions are a serious risk linked to climate heating in northern Scandinavia and highlights the need for regional estimates of the potential for increased $CH_4$ emissions associated with degradation of palsa peatlands.

**Acknowledgements**


We are grateful for the field work support from Mattias Dalkvist, Veronica Escubar Ruiz, and staff at Abisko Scientific Research Station. We also acknowledge that the climate data evaluation has been made by data provided by Abisko Scientific Research Station and the Swedish Infrastructure for Ecosystem Science (SITES). We are grateful to Samuel Valman for advice on data processing.


**Supplementary information**

Supplementary information 1. Excel file containing confusion matrixes for the hard and soft classifications.



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
