# Peer review of "Optical and radar Earth observation data for up-scaling methane emissions linked to permafrost degradation in sub-Arctic peatlands in Northern Sweden"

_Biogeosciences, 2023_

## Author Response (AR1)

Dear Professor Park,

We are very grateful for the support of the editors and reviewers for our work. We have carefully considered all of the reviewers' comments and we have added our response to each point below. I hope that the response is clear, please contact us in case you need any further information from us at this stage.

Yours sincerely

Sofie Sjogersten (on behalf of all the authors)

**Detailed response to the editors and reviewers comments**

**Comments from the editor**
- L 32: CH4 "emissions" from

*We have added this missing word on L31 of the revised MS.*

- L 156: Please briefly describe the measurement principle of the used gas analyzer for readers who are not familiar with the Los Gatos analyzer. Given the importance of the accuracy of your CH4 measurements, it would also be helpful if you provide more details on the employed QA/QC measures, including how you calibrated the analyzer with what standard gases and how you selected the actual data points according to what linearity criteria (e.g., min. r2 values).

*The Los Gatos analyser uses a cavity ring laser spectroscopy system for detecting CO2 and CH4. This technology facilitates parallel detection of CO2 and CH4 in the field. It also have the advantage of being stable over time and not requiring regular calibration. We have confirmed this using standard gases with certified concentrations. Further quality control was carried out to check for instances of ebullition. If there was evidence of ebullition the measure series was excluded from further analysis. We used r2> 0.7 as a cut of for accepting linearity for high CH4 fluxes. We have included more detail about the Los Gatos analyser and the R2 cut off use control on L157-159 which now reads:*

"The Los Gatos analyser uses a cavity ring laser spectroscopy system for detecting $CH_4$ and affords detection of $CH_4$ in the field. It also has the advantage of being stable over time and not requiring regular calibration."

- Discussion: The whole section consists of five paragraphs including the long first one. I thought that the section can be more efficiently structured and strengthened to make the key discussion points more visible. For example, you can use subtitles to emphasize, for instance, the application potentials of your findings, limitations of the extrapolation approach, etc. At least the long first sentence needs to be split.

*We have split the first paragraph in two.*

- Figure 2 (and others): Please use proper axis titles followed by units, such as CH4 emission (mg CH4…). As the reviewers pointed out, you may need to pay more attention to figure details. We have also amened figure 5 and

*We have made this change to Figure 2 and also modified Figure 3 and 4 to address this point from the editor. We also have added north arrows into Figure 5 and 9.*

I would like to ask you to make all the changes easily identifiable in a marked-up manuscript based on your point-by-point responses to the reviewer comments. If possible, please add up your responses to the original reviewer comments and specify the line numbers of the revised parts in your final responses accompanying the revised manuscript.

*We have made all the changes using track changes and explained that changes made in our response to the reviewers below.*

**Reviewer 1**
**General comments:**

The authors present an interesting and novel investigation that attempts to establish a new method for estimating landscape-scale methane emissions from degrading palsa wetlands and for detecting the initial stages of palsa degradation, which represent two urgently pressing issues in ongoing permafrost peatland research. The study combines ultra-high resolution imagery from a UAV with Sentinel-1 and -2 data to extrapolate field-measured methane emissions to the landscape-scale, using relationships with key vegetation types. The authors estimate that for the 50 ha study area, ongoing degradation could increase methane emissions by up to two orders of magnitude, indicating the importance of such observations and measurements for estimating carbon release from palsa sites. The distinction between vegetation types for estimation of CH4 emission rates is much needed for improving our understanding of degradation impacts, as many current studies use more simplistic comparisons between palsas and fens, as the authors note. The introduction, methods and discussion are generally well-written and interesting. I note some points for consideration below.

*We are very happy that the reviewer finds the work important and novel*

However, some further work is required on the analysis and presentation of the measured data before this manuscript is ready for publication. Specifically:

1. The method for calculating the headline finding of increasing CH4 emissions in subsiding areas in these sites (i.e. from 116 kg season$^{-1}$ to 12,960 kg season$^{-1}$) is currently not well explained and unclear to me. Further clarification and justification of this extrapolation analysis is vital to ensure a robust interpretation of these results by other readers.

*For the calculation of the potential change in CH4 emissions in areas that are currently subsiding we combined the InSAR determined subsidence data, the majority classification of the UAV vegetation maps scaled to the 20\*20 m pixels of the InSAR and the CH4 flux data. First we multiplied the subsiding pixels first with the methane emissions for the corresponding vegetation type currently growing in that pixel. There after we assumed we explored the impact of transformation of all of that area to fen vegetation. We have added a detailed explanation of how this was done in the data analysis section. L 282-293 now reads:*

*"We also calculated the net emissions of $CH_4$ across the subsiding parts of the three wetlands (as detected by InSAR). This was done by summing the $CH_4$ emissions attributed to each subsiding InSAR pixel for all three sites based on the dominant vegetation type (using the UAV vegetation classification) in that pixel. To explore how $CH_4$ emissions may change as subsidence progresses and results in palsa collapse, waterlogging and*

*vegetation change we multiplied the currently subsiding wetland areas with the seasonal $CH_4$ emission that was measured in the lower and higher emitting fen vegetation types, respectively to generate two exemplar future emissions scenarios. This "after permafrost loss and palsa collapse" $CH_4$ emission scenario used the assumption that all of the subsiding area will be covered with a particular fen vegetation type. We want to highlight that this is not a realistic scenario as the fen vegetation that will replace the current vegetation on the raised palsa will be a mixture of vegetation communities suited to permanently waterlogged conditions. Hence the calculations of future $CH_4$ emissions should be viewed as illustrations of the ranges of emissions that could occur from the sites following full transition to a fen vegetation type based on their current day emissions."*

2. Secondly, several improvements could be made to assist the readability and presentation of figures and figure captions, most particularly Figure 2 where panel D is currently missing any data.

*Figure 2 had been corrupted during the file upload. We amended the figure to address the problem. We have carried out a detail check of all the figures.*

3. I recommend a global check of the manuscript for typographical errors, because I encountered several mistakes (for example, incorrect figure numbers referred to in the text).

*We have corrected the figure numbers and the symbols that had got corrupted during the upload. In response to the reviewers comments we have carried out a detailed language check as well as carefully checking the MS figure numbering, table formatting etc.*

Overall, this research is a novel and useful contribution to the field of permafrost peatland research, but currently requires some polish to its presentation to be ready for publication.

*We are thankful for to the reviewer for their supportive comments and constructive suggestions for improving the ms. We have outlined in the text below how we have addressed these.*

Specific comments:

L29: It is very unclear both here in the abstract, and in the main text, how the results for future CH4 emissions have been reached. As this is the headline finding of the paper, it is important that further clarification is provided for the methods used to calculate these statistics, and some interpretation in the main text for why there is such a large range between these figures.

*We have described how these calculation where done in the data analysis section as per our response to the reviewers point made above on L282-293.*

L41–43: Although the study's focus is on methane, permafrost thaw can also cause substantial carbon dioxide (CO2) release. It would be useful in the introduction to include some comparison of the relative warming potential or persistence of methane compared to CO2.

*Yes we agree that CO2 fluxes are also important. This is why we had already included the reference to the Olefeldt et al., (2012) study. This paper investigates if permafrost thaw results in overall C loss. Interestingly these authors find that the overall carbon balance in permafrost and non-permafrost peatlands is similar based on their synthesis study (Olefeldt et al., 2012). In response to the reviewers comment we have included some more detailed*

*information about the global warming potential of methane in comparison to CO2 to high light the importance of understanding the fluxes of both these gases as well as their atmospheric residence time. L39-42 now reads: "Any increase in emissions of CH4 is of particular concern, as it is a powerful greenhouse gas with a global warming potential of 28 that of CO2 on a 100 year time frame, with Arctic ecosystems acting as strong emitters of CH4 (Euskirchen et al., 2014; Glagolev et al., 2011; IPCC, 2021; Maksyutov et al., 2010; McGuire et al., 2009; Turetsky et al., 2020)"*

L44-46: Given the current limited mapping of palsas across much of Siberia (e.g. see Fewster et al., 2022), I don't believe there is enough evidence to state a confident areal extent of palsa peatlands across the total permafrost region. Instead, perhaps it would be better to state geographic regions in which they are most commonly found.

*We agree with the reviewer that much uncertainty around this remains and we have removed the quantification as recommended by the reviewer. L43-45 now reads: "They are a common landform in the sporadic and discontinuous zone of the permafrost region where palsa peatlands cover substantial areas of the total permafrost region (Ballantyne C. K., 2018)."*

L46: Tarnocai et al. (2009) do not discuss palsa extents so this reference is unsuitable for making this point.

*We have double checked this and the reviewer is correct and we have removed this reference from this sentence.*

L46-47: These references for peat carbon stocks do not provide estimates for the total carbon stored specifically in palsa mires. Both studies are now > 10 years old and peat carbon estimates been more recently been improved. It seems more appropriate to describe the permafrost peatland carbon store more generally, using updated carbon maps – for example, see Hugelius et al. (2020)

*The data sets in the two papers combined does afford an estimation of palsa carbon so we suggest keeping these. We have added the Hugelius reference to this section as suggested.*

L107: What reference data were used to calculate these MAT and MAP ranges (i.e. weather station observations or gridded climatologies)? A reference to this climate data is important. Please also provide specifics on the time period considered.

*We used data from the Swedish Meterological and Hydrological Institute. In line with the reviewers suggestion, we have add a reference to this in the text as well as details of the data use, e.g. time frames, which was 1990-2020 as requested by the reviewer. See L106*

L130: Figure 1: I would recommend revising the colour scheme used in Figure 1, because the vibrant green and red classes are not colourblind friendly and it is difficult to distinguish the lake shoreline from the terrestrial green classes.

*We have run the map through a colourblind filter and amended it to make it clearer.*

L135-136: This sentence is slightly unclear to me. Were these measurements taken from intact palsa tops? Was there a reason why all vegetation types not studied in both subsiding and intact areas?

*Vegetation changes consistently in response to degradation. In response to the reviewers comment we have specified which vegetation types are associated with the intact drier areas and the waterlogged degraded areas. L133-136 now reads: "Dry lichen, dwarf shrub, moist moss, sphagnum wetland, sedge wetland and willow wetland with the dry lichen, dwarf shrub and moist moss vegetation types occurring on the drier parts on the palsa. The sphagnum wetland, sedge wetland and willow wetland vegetation types occurred in waterlogged fen or thermokarst areas."*

L236-237: How variable were the measured methane emissions within each landcover type? Do the results change substantially if an alternative averaging method is used, e.g. the median?

*The greatest variation in CH4 fluxes were found in the sites with the highest fluxes. The pattern in the data with regards to the magnitude of fluxes from different vegetation types is the same for the arithmetic means as the predicted means of the mixed models. In brief the pattern in the data remains the same when exploring the differences in different ways statistically.*

L236-237: When considering the wider applications of these methods, were the methane measurements from these land covers similar or dissimilar to existing measurements from nearby palsa mires? Do sites require ground truthing of methane measurements before such methods can be employed? Some further comparisons to previous measurements in the discussion would be interesting and useful.

*The range in the CH$_4$ flux measurements in our study compares to fluxes measured in similar systems. Also, in line with data resented in other studies our data show there is high variability in CH$_4$ fluxes among vegetation types, which may be linked to specific species or edaphic conditions that are associated with particular vegetation types. Hence, we anticipate that if the system for upscaling and assessing future methane emission risk we have tested here was to be applied to larger areas in the region or sites further-a-field additional ground methane data as well as vegetation ground truthing to reflect the most relevant communities in that area would be needed. In response to the reviewers point we have added some specific examples of fluxes from other studies to illustrate the range in emissions and how these are associated with vegetation type. L474-487 now reads: "The low emissions from the intact dry palsa areas and trends of increased emission following degradation and flooding were comparable with those reported from other part of Scandinavia and Russia e.g. -1.7 to 1.2 kg CH4 ha-1 season-1 from dry lichen and raised palsa, respectively (Glagolev et al., 2011; Liebner et al., 2015; Miglovets et al., 2021; Nykänen et al., 2003; Olefeldt et al., 2013; Varner et al., 2022). However, there was strong variation in the magnitude of that response depending on the degradation stage, level of flooding and vegetation community present. For example, as long as sites were mainly impacted by increased waterlogging but no vegetation change (i.e. the raised palsa vegetation persisted despite experiencing subsidence), CH4 emissions only increased modestly (Fig 2). This suggest that waterlogging per se was not sufficient to drive high emissions as these were only recorded following flooding and a transition of the vegetation community to Sphagnum mosses, sedges and other wetland adapted species.*

*An important finding in our study is the high CH4 emissions reported from fen and in particular the willow vegetation type. These emissions were consistent across the three study sites and were at the upper range of emissions reported for permafrost fens in the literature e.g. maximum fluxes of ca. 130 to 360 kg CH4 ha-1 and season-1 (Miglovets et al., 2021; Olefeldt et al., 2013) and in the range of emissions from ponds and thermokarst lakes in degrading permafrost in the literature (Dzyuban, 2002; Glagolev et al., 2011; Walter et al., 2006)."*

*We have also clarified the text linked to the reviewers point on the wider applicability in the text, we now highlight that the methods are applicable providing that some location specific ground data are included to support the upscaling. L522-26 now reads: "The bias-correction methods used in this study are simple, effective and transferable, however, there is a need for ground data if these approaches are applied to new areas.Spatially distributed high-resolution studies and data collection in remote areas such as the Canadian Arctic and Siberia are in the context of permafrost degradation and CH4 feedbacks of high value and necessary to calibrate regional to Arctic estimates due to high spatial variability of tundra terrain (Siewert et al., 2021; Siewert & Olofsson, 2020)."*

L278-280: It is unclear what temperature is being measured here – is this soil temperature? Please clarify.

*It is soil temperature; we have added this detail to the text.*

L285: Figure 2: This figure requires revision. Panel D is missing any data bars.

*The missing figure panel must have occurred in the file upload/processing. We have uploaded a new version of Figure 2.*

Standard error whiskers only extend above each bar but should also extend below.

*We have made this change in line with the reviewers suggestion.*

For panel C I would recommend the removal of the dry lichen sub. class as it currently reads as no methane flux, when actually no measurements were recorded.

*We propose to keep the dry lichen sub in the figure to ensure the x-axis are the same format among panels as this facilitates comparison among the different vegetation types and time points. It is clearly stated in the caption that there was no data from the dry lichen sub vegetation type in September.*

In the caption, it is important to clarify where the means and SE are shown (e.g. with bars and whiskers).

*We have already stated in the caption that it is means and SE that are shown in Figure 2.*

L295: Figure 3: Recommend offsetting each data point slightly in the horizontal axis to avoid overwriting.

*We agree with the review about this modification so that it is clear that there is points from all the sites and times. We have made this change to Figure 3.*

L399: Figure 4: Additional information is needed in the caption, detailing what values are shown by boxplot centre lines (medians?), whiskers and closed circles.

*We have added the information in the caption of Figure which now outlines that: Medians, 25 and 75 percentiles are shown. The whiskers show minimum and maximum values excluding outliers, the black dots represent outliers.*

L309: Was there an objective cutoff point for determining a suitable accuracy? How do these performance results compare to previous classification studies?

*There is no universal threshold accuracy target. One that is widely used is 85% overall accuracy but this is for Anderson Level 1 (very broad classes) and a lower accuracy would be expected with more detailed classes such as used in the paper. A cutoff of 70% was used to determine a sufficient accuracy for the vegetation mapping. We decided on this threshold because it is high enough to ensure good overall accuracy, whilst not so high to account for the difficulty of mapping classes with very similar spectral signatures e.g. non-subsiding vegetation classes and their subsiding counterparts. Furthermore, it is also important to note that ground data are never perfect. Indeed, the level of error can be quite large even when arising from authoritative sources. For example, trained aerial photograph interpreters are known to differ in their labelling of forest classes by up to 30% in some studies (Powell et al., 2004). We are being upfront on the quality, which is rare. We have added this justification see L211-216: "A cutoff of 70% was used to determine a sufficient accuracy for the vegetation mapping. We decided on this threshold because it is high enough to ensure good overall accuracy, whilst not so high to account for the difficulty of mapping classes with very similar spectral signatures e.g. non-subsiding vegetation classes and their subsiding counterparts. Furthermore, it is also important to note that ground data are never perfect. Indeed, the level of error can be quite large even when arising from authoritative sources. For example, trained aerial photograph interpreters are known to differ in their labelling of forest classes by up to 30% in some studies (Powell et al., 2004)."*

L330: Does the misclassification of several raised palsa areas in the study area indicate that there may also be an overestimate of predicted emissions in extrapolated results?

*As the variation in CH4 fluxes among the different vegetation types found on the raised palsa areas are low we do not expect this to result in a large over estimation of the predicted CH4 emissions. The fact that the soft classification overestimated the fen vegetation area resulted in an over estimation compared to the UAV data see Table 2 is associated with an over estimation of CH4 fluxes in Table 3 while the bias corrected hard classification performs best relative to the total CH4 emissions estimated from the UAV data. This is the case both for the total emissions and the emissions from the three highest emitting classes.*

L375: Table 3: The caption seems to indicate that four sets of results should be in the table (i.e. the "one model using the UAV data" appears to be missing). The mean and SE should be labelled in the table.

*Yes, one column in Table 3 did not display correctly in the uploaded document. We apologies for this and we have altered the formatting, so this displays correctly. We have specified which column shows the mean and se.*

L383-385: If such a strong bias-correction is required to correct the results for the trained study site, are these methods really suitable for estimating classifications outside of the study domain, where bias-correction would be more difficult without ground-truthed data? Perhaps this evaluation is worth greater discussion.

*The approach presented in this paper, is based on a probability sample of the region mapped. The sample is thus representative of that study area but the referee is correct that it may not be representative of other regions, but we don't go to others in this paper. The methods are transferable, but there is a need for ground data and with that ground data we show the impact of the bias-correction method that is simple but extremely effective. We agree that this point is important. We have brought this out more strongly in the discussion see L522-526.*

L388-389: It is suprising that the hard classification still returned a highly significant correlation to the UAV estimates when the R2 value is much lower than the other

relationships. Does this indicate the low numbers of point estimates (four per model) are unsuitable for significance testing?

*We are grateful for the reviewer to pick up on this. We had made a typo and the hard classification regression model was indeed not significant reflecting the low classification accuracy of the hard classification model. We have updated the text and figure to reflect this.*

L396: Here it would be useful to indicate a range and average for the measured subsidence in mm.

*We agree and have added this detail see L432-434: "Importantly, across the three sites, areas with relatively low levels of ground motion were associated with the highest mean CH4 emissions (F8,1135 = 5.63, p < 0.001, SED = 961.5; Fig 7) while both uplifting (5 to 7.5 mm) and the most subsiding (-10 to -12.5 mm) areas tended to have lower fluxes."*

L413-414: Figure 7 does not have any lettered panels. Should this relate to Figure 9?

*Yes, it should be figure 9. We have corrected this.*

L428-430: This section is very vague, and appears to represent a headline finding for the study. Please clarify in greater detail the vegetation types used to estimate the future emissions, and how these calculations were applied. I did not follow the calculation of these estimates.

*We have explained how this was done in reply to the reviewers earlier point on this issue. We have now added a detailed description of how these emissions were calculated and the assumptions that they were based on in the data analysis section. See L282-293.*

L433: Figure 9: The lettered panel labels are very hard to read. North arrows were only present in one figure panel – please add to all panels.

*We have added north arrows and improved the visibility of the letters.*

L439-441: How comparable are these estimates? If possible, it would be helpful to report a range of estimated values from these studies.

*To enable comparison we have added some fluxes ranges into this section see L474-487.*

L449-453: Similarly, it would be useful to report the ranges to show readers how close these estimates are to the upper end.

*Again, we agree with the reviewer that this would be useful and have added some numbers to illustrate see L474-487.*

L477: Is this referring to the bias-corrected hard classification? It would be helpful to use consistent terms throughout.

*Yes, it is referring to the bias-corrected hard classification. We have checked the text to ensure consistent use of this term.*

L482-484: How easily available are high-resolution remote sensing data for permafrost peatland regions (e.g. central and eastern Siberia are currently very poorly studied)? Some

discussion of the availability of these data would contextualise how easy these methods would be to upscale.

*We have added the following sentence to emphaze this:*

*L523-526: "Spatially distributed high-resolution studies and data collection in remote areas such as the Canadian Arctic and Siberia are in the context of permafrost degradation and $CH_4$ feedbacks of high value and necessary to calibrate regional to Arctic estimates due to high spatial variability of tundra terrain"*

Technical corrections:

*We have made the edits suggested by the reviewer.*

L56: "which have been".

*We have made this change.*

L106: add degree symbol.

*We have made this change.*

L115: Remove additional bracket.

*We have made this change.*

L146: consider removing "point"?

*We have made this change.*

L280: degree symbol missing.

*We have added this.*

L314: Should this read "shadows are not included"?

*Yes, we have corrected this.*

L316: missing symbol before "Sites"

*We have added in the sum symbol.*

L342: change "methane" to CH4?

*We have made this change.*

L344: change "has" to "have"

*We have made this change*

L372: change "were" to "where"

*We have made this change.*

L447: change "were" to "where"

*Were is correct in this instance.*

L477: "Hence improved understanding of the.."

*We have made this change.*

**Reviewer 2**
Review on Capabilities of optical and radar Earth Observation data for up-scaling methane emissions linked to subsidence and permafrost degradation in sub-Arctic peatlands by Sjogersten et al., (bg-2023-17).

This paper investigated the effects of subsidence and permafrost degradation on methane emission over northern Sweden under Arctic warming. Appropriate in-situ measurements and drone & satellite remote sensing data records were used in this investigation. The results show finer spatial resolution vegetation map derived from Sentinel-2 data using a support vector machine classification method. They also found that permafrost degradation and subsidence contributed to methane emission in various ways. They concluded that a fusion of EO data provided an ability to estimate regional methane emissions. The paper covers a topic that is suitable to readers of *Biogeosciences* and should be of particular interest to those interested in climate change impacts on greenhouse gas emissions at higher latitudes.

*We are happy that the reviewer found the paper robustly carried out and of interest to the Biogeosciences audience.*

However, the manuscript has concluded with lack of detail in data (e.g., sentinel-1, -2), methods (e.g., methane emission modeling),

*We are grateful that the reviewer has flagged were further detail could help clarify the approach taken and we have acted on these. We have detailed these changes in response to the reviewers specific comments below.*

and without new findings and strong conclusion (e.g., not clear relationship between permafrost degradation and methane emission). The paper should be suitable for publication following the recommend major revision:

*We dispute that there are no new findings. The testing of the capabilities of Sentinel-2 and Sentinel-1 for estimating methane emissions are novel. Further we show that there are strong links between permafrost degradation and methane emissions and a novel methodology (in ASPIS-InSAR) for assessing the risk of future elevated methane emissions linked to areas that are subsiding and will turn into methane emitting areas in the near future.*

- Line 184-186, pg7: how did authors determine three seasons (spring, summer and autumn) using NDVI and/or air temperature? Was NDVI derived from in-situ camera or satellite? Is air temperature measured at 2-m height? Authors should specify it.

*The length of the three seasons follows a rapid transition due to the high latitude. A typical range was determined estimated using a combination of seasonal NDVI and air temperature data. The NDVI data from the Torneträsk valley is available from a 6-year UAV time-series*

*(partly unpublished), indicating the typical trajectory of rising and falling values during spring and fall of the snow free season and an NDVI peak plateau (summer) station (see (Siewert & Olofsson, 2020). and long-term air temperature data from Abisko is monitored by the Swedish meteorological and hydrological service (www.smhi.se). We have added this detail to indicate how we determined the length of the season to the MS. The details of the UAV derived NDVI camera details is in the indicated references. See L184-190: "A typical range was estimated using a combination of seasonal NDVI and air temperature data. The NDVI data from the Torneträsk valley is available from a 6 year UAV time-series (partly unpublished), indicating the typical trajectory of rising and falling values during spring and fall of the snow free season and an NDVI peak plateau (summer) (see (Siewert & Olofsson, 2020).  Long term air temperature data from Abisko is monitored by the Swedish meteorological and hydrological service (www.smhi.se)."*

- Section 2.4: Authors should include brief description on the UAV image processing (e.g., radiometric calibration for reflectance conversion, observation date and time (e.g, noon?), how many images?), even though specific software and a reference are provided.

*We are happy to offer this detail: Imagery was collected during peak vegetation season in the last week of July and first week of August approximately at noon. The images, ranging between ~1300–2000 per flight, were processed using Pix4D (Berlin, Germany) to produce an orthomosaic for each spectral band. Radiometric calibration was performed using spectral reflectance targets (Mosaic Mill). We have added this detail to the MS on L193-203: "Multispectral and true colour RGB data ultra high resolution (UHR) data were captured across Storflaket and Stordalen in 2020 and from the Tourist St. site in 2021 from ca 106 and 100 m height using a fixed-wing UAV - Sensefly Ebee fitted with a Parrot Sequoia multispectral sensor. Imagery was collected during peak vegetation season in the last week of July and first week of August approximately at noon. The Parrot Sequoia obtained spectral measurements across four bands: Green (550nm ± 40nm), Red (660nm ± 40nm), Red Edge (735nm ± 10nm) and Near Infrared (790nm ± 40nm). The images, ranging between ~1300–2000 per flight, were processed using Pix4D (Berlin, Germany) to produce an orthomosaic for each spectral band. Radiometric calibration was performed using spectral reflectance targets (Mosaic Mill). This resulted in a ground resolution of ~11-13 cm for the multispectral and 2-3 cm for the RGB data. To ensure the orthomosaic and DEM generated from the UAV data were accurately geolocated we collected ca 50 ground control points using a differential GPS (dGPS;Trimble R8s) across all three sites. Further details on the UAV data collection are in de la Barreda Bautista et al. (2022)."*

- Sentinel-2: Optical data are generally contaminated by frequent cloud, aerosol and low solar radiation and angle at higher latitude? Were atmospherically corrected reflectance data used to classify vegetation map? Did you use cloud-free images? What dates did you use? If you combined two or more images for the study domain, did you use the images observed at the same date and time?

*Cloud-free, atmospherically corrected Sentinel-2 reflectance data from 27/07/2019 was used to produce vegetation maps for the sites of interest. Only one image was needed to cover the wider study domain. This has now been added to the text in Section 2.5 see L219-228. "We used a neural network classification approach (Xie et al., 2008) within R using the 'neuralnet' package (Frtisch et al., 2022) to predict vegetation cover over the three sites of interest using Sentinel-2 data (wavebands B2, B3, B4, B5, B6, B7, B8a, B11 and B12) and elevation, slope and roughness data derived from the ArcticDEM, optical-stereo imagery model at 2 m spatial resolution (Morin et al., 2016). Cloud-free Sentinel-2 reflectance data from 27/07/2019 was used to produce vegetation maps for the sites of interest. Reflectance data was atmospherically corrected using Sen2Cor in ESA SNAP. Only one image was required to cover the wider study domain. To match the 20 m × 20 m resolution of the*

*Sentinel-2 pixels, an average elevation was computed for each pixel. Sentinel-2 wavebands 2, 3 and 4 were re-sampled from 10 m to 20 m resolution using nearest neighbour interpolation to match the resolution of the other wavebands. A Sentinel-2 waveband specification is presented in Supplementary information 1."*

-Line209, pg7: Authors should include Sentinel-2 specification, e.g., spatial resolution, temporal resolution, atmospheric correction scheme, wavelength for each band, overpass times and so on. What are wavebands 2., 3, 4, …?

*We have added this information in L219-228. We have included information on atmospheric correction scheme used in the text. A table summarising wavelength and spatial resolution for each Sentinel-2 waveband used in the study has also been included in the supplementary information. We have not included temporal resolution information because this study has only included one acquisition, rather than collecting a time series of data, therefore we did not feel it was relevant. We do specify the data of the image data used.*

-Line241, pg8: in a similar way, authors should include the sensor specification for Sentinel-1 satellite.

*We used C-band SAR data from the Sentinel-1 satellites (European Union's Copernicus Programme) with a wavelength and frequency of 5.55 cm and 5.41 GHz respectively. We have added this information to the revised MS on L257-259.*

- Vegetation classification accuracy assessment from sentinel-2: Accurate vegetation classification map is critical to estimate methane emission in this investigation. However, the 71% accuracy (line 302, pg12) is not high enough to be used as ground-truth data for the Sentinel-2 classification. Authors should justify it.

*This relates to a point made by the other referee. The same response applies here but we also reiterate that ground data are never perfect and a core recommendation in best practice advice is to use ground data that are anticipated to be more accurate than the classification to be assessed. It is also important to not imply the ground data are perfect hence why the paper explicitly states the accuracy (many studies do not). We have outline this in the text on see L211-216.*

- Authors have not included how to upscale methane emission from in-situ. Authors should provide the details on modeling methane emission in method section.

*We have described how this was done in our response to reviewer 1, see L282-293.*

-There is not clear new findings compared to previous studies. Generally higher water content, the higher methane emission.

*We disagree that there is no novel findings. The novelty in this work is the importance of higher resolution vegetation classification based on detail field data using a combination of high-resolution UAV data, Sentinel- 1 and 2 to map methane emissions. We have also provided proof of concept of a system (i.e. the InSAR subsidence data) for predicting future areas at risk of becoming high methane emitters. This work has considerable importance for quantifying the regional emissions of CH4 from these types of systems and also forecasting of CH4 emissions from degrading palsa peatlands.*

- They concluded a fusion of EO data types provides the ability to estimate methane emission. Authors have not included how to fuse EO data records for regional methane emission. Authors should revise conclusions in a better way.

*We did not mean data fusion as semantically used within the field of remote sensing. Combining is a more appropriate word to use in this context. We have made this change of words in the revision on L29.*

Additional edits are noted below:

*We are thankful for the reviewer to provide these detail suggestions for additions and clarification to the MS text. We have addressed all the points as listed below.*

Line 2-3, pg1: This investigation focuses on Northern Sweden. Sub-Arctic peatlands is not clear to describe the study domain. Authors should include study area into the title, e.g., over Northern Sweden.

*We have added this to the title.*

Line 47, pg2: which data? Is it observation or model simulation? Authors should specify it. Is the 100 GtC stored in entire Arctic region or study area? Authors should clarify it.

*We have specified it is for the Arctic region on L46. The Tarnocai paper uses peat cores from across the Arctic for its estimates and modelled permafrost data and remotely sensed vegetation data.*

Line65-66, pg3: It seems to me that there are too many references (8 papers). Authors might make a few separate statements, by grouping based on subsidence detection algorithm, sensor types.

*We have streamlined the referencing in this section. Now the references are directly relevant.*

Line69, pg3: Publically should be better instead of freely.

*We have made this change.*

Line75-77, pg3: is it underestimated all the time? Authors should include some references to support this statement.

*We have added a reference that discusses the concept of hotspots in biogeochemistry to L75*

Line77-79, pg3: Authors should include relevant papers to support this statement.

*We have added a reference to support this statement.*

Line83-87, pg3: Authors should include relevant papers to support this statement.

*We have added a reference to support this statement.*

Line112-115, pg4: how did you identify vegetation species? By visual inspection?

*Yes, this was done by a team member with extensive experience of working with Arctic wetland plants.*

Line194-196, pg7: This statement is not clear to me. Did you use DEM measured from Parrot Sequoia multispectral camera?

*Yes, that is correct. We have clarified this I the text.*

Line197, pg7: Authors should provide the resampling method.

*The resampling method (nearest neighbour interpolation) has now been included on L226*

Line201-205, pg7: Did authors use UAV reflectance data to apply support vector machine classification (ArcGIS)? Which bands did you use? Authors should provide more details on the classification method.

*The four bands described in the previous paragraph were used for SVM classification. This has now been clarified in the description on L196-197.*

Line208, pg7: Are there any specific reasons why authors used a different classification algorithm (neural network classification) and tools (QGIS) from the UAV based-classification?

*Due to the coarser Sentinel-2 spatial resolution (20m) compared to UAV spatial resolution (0.5m), we wanted to use a method that would enable fuzzy classification using proportions of each UAV land cover class within each Sentinel-2 pixel. Neural networks in R were more easily implemented for such purposes, with the benefit of faster processing speeds. We have now added this detail to Section 2.5, as well as corrected the tool used to build the neural network (manuscript previously said QGIS) see L233-234.*

Line241, pg8: Authors should include the frequency for C-band.

*Wavelength and frequency of the Sentinel-1 C-band instrument has been added on L257-259*

Line245, pg8: What is APSIS or IBSAS?

*ASPIS stands for Advanced Pixel System using Intermittent Baseline Subset. This is set out in the introduction. We have now written out what the ISBAS (Intermittent Small Baseline Subset) abbreviation stands for in the text on L64*

Line271, pg9: how did authors identify signs of permafrost degradation/subsidence? Was it done from fieldwork by visual inspection? Authors should clarify it.

*Subsiding areas was identified visually based on changes in topography, subsidence, and waterlogging. We have clarified this in the experimental design section on L136-138.*

Line278-279, pg9: In contrast to temperature, soil moisture does not show large variation from June to September, except for Dry lichen sub and moist moss sub. Dry areas get drier as temperature increases. What is initial thaw period (e.g., snow melting, soil thawing)? Did you define spring thaw timing in the previous section?

*The raised parts of the sites tend to get snow free in April/May but snow can remain in hollows into June. We have added this detail to the site description on L110.*

Line280, pg9: Figure 3b shows lowest temperature occurs in September, not in June.

*We have corrected this mistake.*

Line287, pg10: What does season mean? Did you define the season?

*We used climate data and growth indicators to identify the period during which we would expect microbial activity, this is what we here refer to as season. We identify June as spring, July-Aug as summer and September as autumn. We have added this detail to the paper on L184-190.*

Line398, pg18: Fig 6 should be fig 7.

*We have corrected this.*

Line400, pg18: Fig 7 should be fig 8.

*We have corrected this.*

Line443, pg22: did you present the degradation stage and level of flooding in the results? If not, authors should revise it.

*We have clarified this sentence to make it clear what we refer to when we talk about degradation on L480-482.*

Line454-455, pg22: Are these species in wetland vegetation shown in Table 3? Or were these species found in the study domain? Authors should clarify it.

*We have specified that these species form part of the vegetation communities in the study area on L489-490.*

Line464-466, pg22: Are there any references on species composition impacts on methane emissions?

*We have added references to support this on L504-506.*

Figure 4: Authors should describe what error bars and black dots represent.

*We have added this explanation in the figure caption.*

Figure 9: what background map did you use? Google earth map? Authors should describe it in the caption.

*We have clarified in the figure caption that the backgrounds are orthophotos over the sites.*

Table 3: What are three different models? What is a model using UAV data? Authors should describe in main text.

*The models are described in section 2.5 Sentinel-2 remotely sensed data for vegetation mapping. Unfortunately, a section of the table with the UAV had got cut off in the PDF, we*

*have amended this now. We have improved the description of how this data was derived in the table heading and refers to Table 3 in the section of the text where we describe the results of the methane scaling using the different Selntinel-2 models compared to the UAV data methane estimates.*

---

## Author Response (AR2)

Sutton Bonington 17 July 2023

Dear Dr Park,

Thank you for your positive response to our revisions. We have now addressed the second set of reviewers comments. The specific responses are highlighted below.

Yours sincerely

Sofie Sjogersten (on behalf of all the authors).

Detailed response

Reviewer 1

1. Despite the authors acknowledging that Figure was corrupted on submission, they have replaced the figure with the same version (i.e., still missing data points for panel D in the pdfs I received). This must be ammended before the paper is finalised.

*We have ensured this figure now displays correctly.*

2. Similarly, the authors agreed with me that Figure 3 could be improved by slightly offsetting the data points in the horizontal axis, but on both the tracked changes and manuscript files I can't see any difference in the revised version of this Figure. I suggest the authors may have misplaced the revised figure?

*We had made a small displacement in the figure in the resubmitted MS. As the reviewer did not notice this we now increased the displacement so that it is clearer to the reader.*

3. Minor grammatical point: On the author's correction to L133-136 Sphagnum requires capitalising and italicising in the revised manuscript.

*We have corrected this.*

4. Minor grammatical point: In the author's correction to L39-42 should it read "28 times that of CO2"? If so this should be amended.

*Thank you, we have made this change.*

Once these final changes are made, and the Figures are replaced with the correct versions, I believe the paper will be ready for acceptance.

Reviewer 2

The authors have adequately addressed most of the issues raised by the reviewers, but some minor revisions could be addressed for publication.

Minor comments:
Authors should add the novel methodology described in their response to the discussion section, if it is not included; "a novel methodology (in ASPIS-InSAR) for assessing the risk of future elevated methane emissions linked to areas that are subsiding and will turn into methane emitting areas in the near future."

*We have reworded the sentence in the discussion that addresses this point following the reviewers suggestion on L557.*

Line 184-186, pg7: It is still unclear. It seems to me that authors used vegetation phenology metrics (e.g., onset, offset) derived from seasonal NDVI variation and air temperature. If so, authors should provide a brief description on which phenology detection method has been used. For the air temperature, did authors use 0 degree temperature as a threshold to identify vegetation phenology onset? Please specify it. The methodology used in the paper should be repeatable.

*We have added details on the approach we took to the methods section. L190 and 199 now reads: We defined the spring period to start from when the ground became snow free, air temperature rose above 5°C and soil temperatures at 5 cm depth rose to above zero while NDVI remained low (< 0.25), this coincided with the month of June. We defined the summer growing season as the periods from the start of July when the NDVI increased from the non-growing season NDVI of < 0.25 to > 0.5 and soil temperatures were consistently over 5°C and daily mean air temperatures were mostly 10°C or above. The NDVI peaked mid-August with values around 0.7. The summer period included July and August. We identified the onset of autumn by a drop in NDVI at the end of August that occurred in parallel with a drop in the daily mean air temperature which fell below 10°C at this time. By the end of September, the vegetation had senesced, and the mean daily air temperatures started reaching freezing conditions.*

Section 2.4: Did authors use Pix4D mapper? There are a series of the Pix4D software. Please specify it.

*We used version 4.6.4. We have added this to the paper.*

---

## Author Response (AR3)

Sutton Bonington 2023-08-01

Dear Biogeosciences editorial team,

I am very pleased that our paper is now ready for production. I have uploaded the required files and I look forward to seeing the final paper in print. Please let me know if you require any further information.

Best wishes

Sofie Sjogersten